# BENCHMARKING CONSTRAINT INFERENCE IN INVERSE REINFORCEMENT LEARNING

**Guiliang Liu[1,2,3], Yudong Luo[2,3], Ashish Gaurav[2,3], Kasra Rezaee[4], Pascal Poupart[2,3]**
[1]The Chinese University of Hong Kong, Shenzhen, [2]University of Waterloo, [3]Vector Institute, [4]Huawei
`liuguiliang@cuhk.edu.cn, yudong.luo@uwaterloo.ca,`
`ashish.gaurav@uwaterloo.ca,kasra.rezaee@huawei.com,ppoupart@uwaterloo.ca`

## ABSTRACT

When deploying Reinforcement Learning (RL) agents into a physical system, we must ensure that these agents are well aware of the underlying constraints. In many real-world problems, however, the constraints are often hard to specify mathematically and unknown to the RL agents. To tackle these issues, Inverse Constrained Reinforcement Learning (ICRL) empirically estimates constraints from expert demonstrations. As an emerging research topic, ICRL does not have common benchmarks, and previous works tested algorithms under hand-crafted environments with manually-generated expert demonstrations. In this paper, we construct an ICRL benchmark in the context of RL application domains, including robot control, and autonomous driving. For each environment, we design relevant constraints and train expert agents to generate demonstration data. Besides, unlike existing baselines that learn a "point estimate" constraint, we propose a variational ICRL method to model a posterior distribution of candidate constraints. We conduct extensive experiments on these algorithms under our benchmark and show how they can facilitate studying important research challenges for ICRL. The benchmark, including the instructions for reproducing ICRL algorithms, is available at `https://github.com/Guiliang/ICRL-benchmarks-public`.

## 1 INTRODUCTION

Constrained Reinforcement Learning (CRL) typically learns a policy under some known or predefined constraints (Liu et al., 2021). This setting, however, is not realistic in many real-world problems since it is difficult to specify the exact constraints that an agent should follow, especially when these constraints are time-varying, context-dependent, and inherent to experts' own experience. Further, such information may not be completely revealed to the agent. For example, human drivers tend to determine an implicit speed limit and a minimum gap to other cars based on the traffic conditions, rules of the road, weather, and social norms. To derive a driving policy that matches human performance, an autonomous agent needs to infer these constraints from expert demonstrations.

An important approach to recovering the underlying constraints is Inverse Constrained Reinforcement Learning (ICRL) (Malik et al., 2021). ICRL infers a constraint function to approximate constraints respected by expert demonstrations. This is often done by alternating between updating an imitating policy and a constraint function. Figure 1 summarizes the main procedure of ICRL. As an emerging research topic, ICRL does not have common datasets and benchmarks for evaluation. Existing validation methods heavily depend on the safe-Gym (Ray et al.,

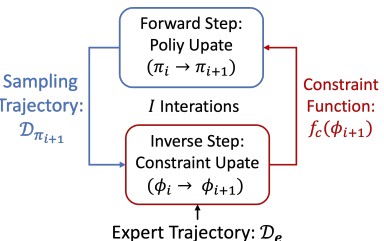

Figure 1: The flowchart of ICRL.

2019) environments. Utilizing these environments has some important drawbacks: 1) These environments are designed for control instead of constraint inference. To fill this gap, previous works often pick some environments and add external constraints to them. Striving for simplicity, many of the selected environments are deterministic with discretized state and action spaces (Scobee & Sastry, 2020; McPherson et al., 2021; Glazier et al., 2021; Papadimitriou et al., 2021; Gaurav et al., 2022). Generalizing model performance in these simple environments to practical applications is difficult.

2) ICRL algorithms require expert demonstrations respecting the added constraints while general RL environments do not include such data, and thus previous works often manually generate the expert data. However, without carefully fine-tuning the generator, it is often unclear how the quality of expert trajectories influences the performance of ICRL algorithms.

In this paper, we propose a benchmark for evaluating ICRL algorithms. This benchmark includes a rich collection of testbeds, including virtual, realistic, and discretized environments. The virtual environments are based on MuJoCo (Todorov et al., 2012), but we update some of these robot control tasks by adding location constraints and modifying dynamic functions. The realistic environments are constructed based on a highway vehicle tracking dataset (Krajewski et al., 2018), so the environments can suitably reflect what happens in a realistic driving scenario, where we consider constraints about car velocities and distances. The discretized environments are based on grid-worlds for visualizing the recovered constraints (see Appendix B). To generate the demonstration dataset for these environments, we expand the Proximal Policy Optimization (PPO) (Schulman et al., 2017) and policy iteration (Sutton & Barto, 2018) methods by incorporating ground-truth constraints into the optimization with Lagrange multipliers. We empirically demonstrate the performance of the expert models trained by these methods and show the approach to generating expert demonstrations.

For ease of comparison, our benchmark includes ICRL baselines. Existing baselines learn a constraint function that is most likely to differentiate expert trajectories from the generated ones. However, this point estimate (i.e., single constraint estimate) may be inaccurate. On the other hand, a more conceptually-satisfying method is accounting for all possibilities of the learned constraint by modeling its posterior distribution. To extend this Bayesian approach to solve the task in our benchmark, we propose a Variational Inverse Constrained Reinforcement Learning (VICRL) algorithm that can efficiently infer constraints from the environment with a high-dimensional and continuous state space.

Besides the above regular evaluations, our benchmark can facilitate answering a series of important research questions by studying how well ICRL algorithms perform 1) when the expert demonstrations may *violate* constraints (Section 4.3) 2) under *stochastic* environments (Section 4.4) 3) under environments with *multiple* constraints (Section 5.2) and 4) when recovering the exact *least constraining* constraint (Appendix B.2).

## 2 BACKGROUND

In this section, we introduce Inverse Constrained Reinforcement Learning (ICRL) that alternatively solves both a forward Constrained Reinforcement Learning problem (CRL) and an inverse constraint inference problem (see Figure 1).

### 2.1 CONSTRAINED REINFORCEMENT LEARNING

Constrained Reinforcement Learning (CRL) is based on Constrained Markov Decision Processes (CMDPs) $\mathcal{M}^c$, which can be defined by a tuple $(\mathcal{S}, \mathcal{A}, p_\mathcal{R}, p_\mathcal{T}, \{(p_{\mathcal{C}_i}, \epsilon_i)\}_{\forall i}, \gamma, T)$ where: 1) $\mathcal{S}$ and $\mathcal{A}$ denote the space of states and actions. 2) $p_\mathcal{T}(s'|s, a)$ and $p_\mathcal{R}(r|s, a)$ define the transition and reward distributions. 3) $p_{\mathcal{C}_i}(c|s, a)$ denotes a stochastic constraint function with an associated bound $\epsilon_i$, where $i$ indicates the index of a constraint, and the cost $c \in [0, \infty]$. 4) $\gamma \in [0, 1)$ is the discount factor and $T$ is the planning horizon. Based on CMDPs, we define a trajectory $\tau = [s_0, a_0, ..., a_{T-1}, s_T]$ and $p(\tau) = p(s_0) \prod_{t=0}^{T-1} \pi(a_t|s_t) p_\mathcal{T}(s_{t+1}|s_t, a_t)$. To learn a policy under CMDPs, CRL agents commonly consider the following optimization problems.

**Cumulative Constraints.** We consider a CRL problem that finds a policy $\pi$ to maximize expected discounted rewards under a set of cumulative soft constraints:

$$\arg\max_\pi \mathbb{E}_{p_\mathcal{R}, p_\mathcal{T}, \pi} \left[ \sum_{t=0}^{T} \gamma^t r_t \right] + \frac{1}{\beta} \mathcal{H}(\pi) \text{ s.t. } \mathbb{E}_{p_{\mathcal{C}_i}, p_\mathcal{T}, \pi} \left[ \sum_{t=0}^{T} \gamma^t c_i(s_t, a_t) \right] \le \epsilon_i \ \forall i \in [0, I] \quad (1)$$

where $\mathcal{H}(\pi)$ denotes the policy entropy weighted by $\frac{1}{\beta}$. This formulation is useful given an infinite horizon ($T = \infty$), where the constraints consist of bounds on the expectation of cumulative constraint values. In practice, we commonly use this setting to define *soft* constraints since the agent can recover from an undesirable movement (corresponding to a high cost $c_i(s_t, a_t)$) as long as the discounted additive cost is smaller than the threshold ($\epsilon_i$).

**Trajectory-based Constraints.** An alternative approach is directly defining constraints on the sampled trajectories without relying on the discounted factor:

$$\arg\max_{\pi} \mathbb{E}_{p_{\mathcal{R}}, p_{\mathcal{T}}, \pi} \left[ \sum_{t=0}^{T} \gamma^t r_t \right] + \frac{1}{\beta}\mathcal{H}(\pi) \text{ s.t. } \mathbb{E}_{\tau \sim (p_{\mathcal{T}}, \pi), p_{\mathcal{C}_i}} \left[ c_i(\tau) \right] \leq \epsilon_i \; \forall i \in [0, I] \quad (2)$$

Depending on how we define the trajectory cost $c(\tau)$, the trajectory constraint can be more restrictive than the cumulative constraint. For example, inspired by Malik et al. (2021), we define $c(\tau) = 1 - \prod_{(s,a)\in\tau} \phi(s, a)$ where $\phi(s, a)$ indicates the probability that performing action $a$ under a state $s$ is safe (i.e., within the support of the distribution of expert demonstration). Compared to the above additive cost, this factored cost imposes a stricter requirement on the safety of each state-action pair in a trajectory (i.e., if $\exists (\bar{s}, \bar{a}) \in \tau, \phi(\bar{s}, \bar{a}) \to 0$, then $\prod_{(s,a)\in\tau} \phi(\cdot) \to 0$ and thus $c(\tau) \to 1$).

## 2.2 INVERSE CONSTRAINT INFERENCE

In practice, instead of observing the constraint signals, we often have access to expert demonstrations that follow the underlying constraints. Under this setting, the agent must recover the constraint models from the dataset. This is a challenging task since there might be various equivalent combinations of reward distributions and constraints that can explain the same expert demonstrations (Ziebart et al., 2008). To guarantee the identifiability, ICRL algorithms generally assume that rewards are observable, and the goal is to *recover the minimum constraint set that best explains the expert data* (Scobee & Sastry, 2020). This is the key difference with Inverse Reinforcement Learning (IRL), which aims to learn rewards from an unconstrained MDP.

**Maximum Entropy Constraint Inference.** Existing ICRL works commonly follow the Maximum Entropy framework. The likelihood function is represented as follow (Malik et al., 2021):

$$p(\mathcal{D}_e | \phi) = \frac{1}{(Z_{\mathcal{M}^{\hat{c}_\phi}})^N} \prod_{i=1}^{N} \exp\left[r(\tau^{(i)})\right] \mathbb{1}^{\mathcal{M}^{\hat{c}_\phi}}(\tau^{(i)}) \quad (3)$$

where 1) $N$ denotes the number of trajectories in the demonstration dataset $\mathcal{D}_e$, 2) the normalizing term $Z_{\mathcal{M}^{\hat{c}_\phi}} = \int \exp\left[r(\tau)\right] \mathbb{1}^{\mathcal{M}^{\hat{c}_\phi}}(\tau) \mathrm{d}\tau$, and 3) the indicator $\mathbb{1}^{\mathcal{M}^{\hat{c}_\phi}}(\tau^{(i)})$ can be defined by $\phi(\tau^{(i)}) = \prod_{t=1}^{T} \phi_t$ and $\phi_t(s_t^i, a_t^i)$ defines to what extent the trajectory $\tau^{(i)}$ is feasible, which can substitute the indicator in Equation (3), and thus we define:

$$\log\left[p(\mathcal{D}_e | \phi)\right] = \sum_{i=1}^{N} \left[r(\tau^{(i)}) + \log \prod_{t=0}^{T} \phi_\theta(s_t^{(i)}, a_t^{(i)})\right] - N \log \int \exp[r(\hat{\tau})] \prod_{t=0}^{T} \phi_\theta(\hat{s}_t, \hat{a}_t) \mathrm{d}\hat{\tau} \quad (4)$$

We can update the parameters $\theta$ of the feasibility function $\phi$ by computing the gradient of this likelihood function:

$$\nabla_\theta \log\left[p(\mathcal{D}_e | \phi)\right] = \sum_{i=1}^{N} \left[\nabla_\phi \sum_{t=0}^{T} \log[\phi_\theta(s_t^{(i)}, a_t^{(i)})]\right] - N\mathbb{E}_{\hat{\tau} \sim \pi_{\mathcal{M}^\phi}} \left[\nabla_\phi \sum_{t=0}^{T} \log[\phi_\theta(\hat{s}_t, \hat{a}_t)]\right] \quad (5)$$

where $\hat{\tau}$ is sampled based on executing policy $\pi_{\mathcal{M}^{\hat{\phi}}}(\hat{\tau}) = \frac{\exp[r(\hat{\tau})]\phi(\hat{\tau})}{\int \exp[r(\tau)]\phi(\tau)\mathrm{d}\tau}$. This is a maximum entropy policy that can maximize cumulative rewards subject to $\pi_{\mathcal{M}^\phi}(\tau) = 0$ when $\sum_{(s,a)\in\tau} \hat{c}_\phi(s, a) > \epsilon$ (note that $\hat{c}_\phi(s, a) = 1 - \phi_t$ as defined above). In practice, we can learn this policy by constrained maximum entropy RL according to objective (2. In this sense, ICRL can be formulated as a bi-level optimization problem that iteratively updates the upper-level objective (2) for policy optimization and the lower-level objective (5) for constraint learning until convergence ($\pi$ matches the expert policy).

## 3 EVALUATION METHODS

In this section, we introduce our approach to evaluating the ICRL algorithms. To quantify the performance of ICRL algorithms, the benchmark must be capable of determining whether the learned constraint is correct. However, the *true* constraints satisfied by real-world agents are often unavailable, for example, the exact constraints satisfied by human drivers are unknown, and we thus cannot use real-world datasets as expert demonstrations. To solve these issues, our ICRL benchmark enables incorporating *external constraints* into the environments and *generates expert demonstrations* satisfying these constraints. Based on the dataset, we design evaluation metrics and baseline models for comparing ICRL algorithms under our benchmark.

## 3.1 DEMONSTRATION GENERATION

To generate the dataset, we train a PPO-Lagrange (PPO-Lag) under the CMDP with the known constraints (Table 1 and Table 3) by performing the following steps:

**Training Expert Agent.** We train expert agents by assuming the ground-truth constraints are unknown under different environments (introduced in Appendix B, Section 4 and Section 5). The cost function $c^*(s_t, a_t)$ returns 1 if the constraint is violated when the agent performs $a_t$ in the state $s_t$ otherwise 0. In the environments (in Section 4 and Section 5) with continuous state and action spaces, we train the expert agent by utilizing the Proximal Policy Optimization Lagrange (PPO-Lag) method in Algorithm 1. In the environment with discrete action and state space, we learn the expert policy with the Policy Iteration Lagrange (PI-Lag) method in Algorithm 2. The empirical results (Figure D.1 and Figure 6) show that PI-Lag and PPO-Lag can achieve satisfactory performance given the ground-truth constraint function.

**Generating a Dataset with Expert Agents.** We initialize $\mathcal{D}_e = \{\emptyset\}$ and run the trained expert agents in the testing environments. While running, we monitor whether the ground-truth constraints are violated until the game ends. If yes, we mark this trajectory as infeasible, otherwise, we record the corresponding trajectory: $\mathcal{D}_e = \mathcal{D}_e \cup \{\tau_e\}$. We repeat this process until the demonstration dataset has enough trajectories. To understand how $\mathcal{D}_e$ influences constraint inference, our benchmark enables studying the option of including these infeasible trajectories in the expert dataset (Section 4.3). Note there is no guarantee the trajectories in $\mathcal{D}_e$ are optimal in terms of maximizing the rewards. For more details, please check Appendix E. Our experiment (Section 4.2) shows ICRL algorithms can outperform PPO-Lag under some easier environments.

## 3.2 BASELINES

For ease of comparison, our benchmark contains the following state-of-the-art baselines:

**Binary Classifier Constraint Learning (BC2L)** build a binary classifier to differentiate expert trajectories from the generated ones to solve the constraint learning problem and utilizes PPO-Lag or PI-Lag (Algorithms 1 and 2) to optimize the policy given the learned constraint. BC2L is *independent* of the maximum entropy framework, which often induces a loss of identifiability in the learned constraint models.

**Generative Adversarial Constraint Learning (GACL)** follows the design of Generative Adversarial Imitation Learning (GAIL) (Ho & Ermon, 2016), where $\zeta(s, a)$ assigns 0 to violating state-action pairs and 1 to satisfying ones. In order to include the learned constraints into the policy update, we construct a new reward $r'(s, a) = r(\cdot) + \log[\zeta(\cdot)]$. In this way, GAIL enforces hard constraints by directly punishing the rewards on the violating states or actions through assigning them $-\infty$ penalties (without relying on any constrained optimization technique).

**Maximum Entropy Constraint Learning (MECL)** is based on the maximum entropy IRL framework (Ziebart et al., 2008), with which Scobee & Sastry (2020) proposed an algorithm to search for constraints that most increase the likelihood of observing expert demonstrations. This algorithm focused on discrete state spaces only. A following work (Malik et al., 2021) expanded MECL to continuous states and actions. MECL utilizes PPO-Lag (or PI-Lag in discrete environments) to optimize the policy given the learned constraint.

**Variational Inverse Constrained Reinforcement Learning (VICRL)** is also based on the maximum entropy IRL framework (Ziebart et al., 2008), but instead of learning a "point estimate" cost function, we propose inferring the distribution of constraint for capturing the epistemic uncertainty in the demonstration dataset. To achieve this goal, VICRL infers the distribution of a feasibility variable $\Phi$ so that $p(\phi|s, a)$ measures to what extent an action $a$ should be allowed in a particular state $s$[1]. The instance $\phi$ can define a soft constraint given by: $\hat{c}_\phi(s, a) = 1 - \phi$ where $\phi \sim p(\cdot|s, a)$. Since $\Phi$ is a continuous variable with range $[0, 1]$, we parameterize $p(\phi|s, a)$ by a Beta distribution:

$$\phi(s, a) \sim p(\phi|s, a) = \text{Beta}(\alpha, \beta) \text{ where } [\alpha, \beta] = \log[1 + \exp(f(s, a))] \tag{6}$$

here $f$ is implemented by a multi-layer network with 2-dimensional outputs (for $\alpha$ and $\beta$). In practice, the true posterior $p(\phi|\mathcal{D}_e)$ is intractable for high-dimensional input spaces, so VICRL learns

---

[1]We use a uppercase letter and a lowercase letter to define a random variable and an instance of this variable.

an approximate posterior $q(\phi|\mathcal{D}_e)$ by minimizing $\mathcal{D}_{kl}\Big[q(\phi|\mathcal{D}_e)\|p(\phi|\mathcal{D}_e)\Big]$. This is equivalent to maximizing an Evidence Lower Bound (ELBo):

$$\mathbb{E}_q\Big[\log p(\mathcal{D}_e|\phi)\Big] - \mathcal{D}_{kl}\Big[q(\phi|\mathcal{D}_e)\|p(\phi)\Big] \tag{7}$$

where the *log-likelihood* term $\log p(\mathcal{D}_e|\phi)$ follows Equation 3 and the major challenge is to define the *KL divergence*. Striving for the ease of computing mini-batch gradients, we approximate $\mathcal{D}_{kl}\Big[q(\phi|\mathcal{D})\|p(\phi)\Big]$ with $\sum_{(s,a)\in\mathcal{D}}\mathcal{D}_{kl}\Big[q(\phi|s,a)\|p(\phi)\Big]$. Since both the posterior and the prior are Beta distributed, we define the KL divergence by following the Dirichlet VAE Joo et al. (2020):

$$\begin{aligned}
\mathcal{D}_{kl}\Big[q(\phi|s,a)\|p(\phi)\Big] = {} & \log\Big(\frac{\Gamma(\alpha+\beta)}{\Gamma(\alpha^0+\beta^0)}\Big) + \log\Big(\frac{\Gamma(\alpha^0)\Gamma(\beta^0)}{\Gamma(\alpha)\Gamma(\beta)}\Big) \\
& + (\alpha-\alpha^0)\Big[\psi(\alpha)-\psi(\alpha+\beta)\Big] + (\beta-\beta^0)\Big[\psi(\beta)-\psi(\alpha+\beta)\Big]
\end{aligned} \tag{8}$$

where 1) $[\alpha^0,\beta^0]$ and $[\alpha,\beta]$ are parameters from the prior and 2) the posterior functions and $\Gamma$ and $\psi$ denote the gamma and the digamma functions. Note that the goal of ICRL is to infer the least constraining constraint for explaining expert behaviors (see Section 2.2). To achieve this, previous methods often use a regularizer $\mathbb{E}[1-\phi(\tau)]$ Malik et al. (2021) for punishing the scale of constraints, whereas our KL-divergence extends it by further regularizing the variances of constraints.

### 3.3 Experiment Setting

**Running Setting.** Following Malik et al. (2021), we evaluate the quality of a recovered constraint by checking if the corresponding imitation policy can maximize the cumulative rewards with a minimum violation rate for the ground-truth constraints. We repeat each experiment with different random seeds, according to which we report the mean $\pm$ standard deviation (std) results for each studied baseline and environment. For the details of model parameters and random seeds, please see Appendix C.3.

**Evaluation Metric.** To be consistent with the goal of ICRL, our benchmark uses the following evaluation metrics to evaluate the tasks 1) *constraint violation rate* quantifies the probability with which a policy violates a constraint in a trajectory. 2) *Feasible Cumulative Rewards* computes the total number of rewards that the agent collects before violating any constraint.

## 4 Virtual Environment

An important application of RL is robotic control, and our virtual benchmark mainly studies the robot control task with a location constraint. In practice, this type of constraint captures the locations of obstacles in the environment. For example, the agent observes that none of the expert agents visited some places. Then it is reasonable to infer that these locations must be unsafe, which can be represented by constraints. Although the real-world tasks might require more complicated constraints, our benchmark, as the first benchmark for ICRL, could serve as a stepping stone for these tasks.

### 4.1 Environment Settings

We implement our virtual environments by utilizing MuJoCo Todorov et al. (2012), a virtual simulator suited to robotic control tasks. To extend MuJoCo for constraint inference, we modify the MuJoCo environments by incorporating some predefined constraints into each environment and adjusting some reward terms. Table 1 summarizes the environment settings (see Appendix C.1 for more details). The virtual environments have 5 different robotic control environments simulated by MuJoCo. We add constraints on the $X$-coordinate of these robots: 1) For the environments where it is relatively easier for the robot to move backward rather than forward (e.g., Half-Cheetah, Ant, and Walker), our constraints bound the robot in the forward direction (the $X$-coordinate having positive values), 2) For the environments where moving forward is easier (e.g., Swimmer), the constraints bound the robot in the backward direction (the $X$-coordinate having negative values). In these environments, the rewards are determined by the distance that a robot moves between two continuous time steps, so the robot is likely to violate the constraints in order to maximize the magnitude of total rewards (see our analysis below). To increase difficulty, we include a Biased Pendulum environment that has a larger reward on the left side. We nevertheless enforce a constraint to prevent the agent to go too far on the left side. The agent must resist the influence of high rewards and stay in safe regions.

Table 1: The virtual and realistic environments in our benchmark.

| Type | Name | Dynamics | Obs. Dim. | Act. Dim. | Constraints |
|------|------|----------|-----------|-----------|-------------|
| | Blocked Half-cheetah | Deterministic | 18 | 6 | X-Coordinate $\geq$ -3 |
| | Blocked Ant | Deterministic | 113 | 8 | X-Coordinate $\geq$ -3 |
| Virtual | Biased Pendulumn | Deterministic | 4 | 1 | X-Coordinate $\geq$ -0.015 |
| | Blocked Walker | Deterministic | 18 | 6 | X-Coordinate $\geq$ -3 |
| | Blocked Swimmer | Deterministic | 10 | 2 | X-Coordinate $\leq$ 0.5 |

**The significance of added Constraints.** The thresholds of the constraints in Table 1 are determined experimentally to ensure that these constraints "matter" for solving the control problems. This is shown in Figure D.1 in the appendix: 1) without knowing the constraints, a PPO agent tends to violate these constraints in order to collect more rewards within a limited number of time steps. 2) When we inform the agent of the ground-truth constraints (with the Lagrange method in Section 3.1), the PPO-Lag agent learns how to stay in the safe region, but the scale of cumulative rewards is likely to be compromised. Based on these observations, we can evaluate whether the ICRL algorithms have learned a satisfying constraint function by checking whether the corresponding RL agent can gather more rewards by performing feasible actions under the safe states.

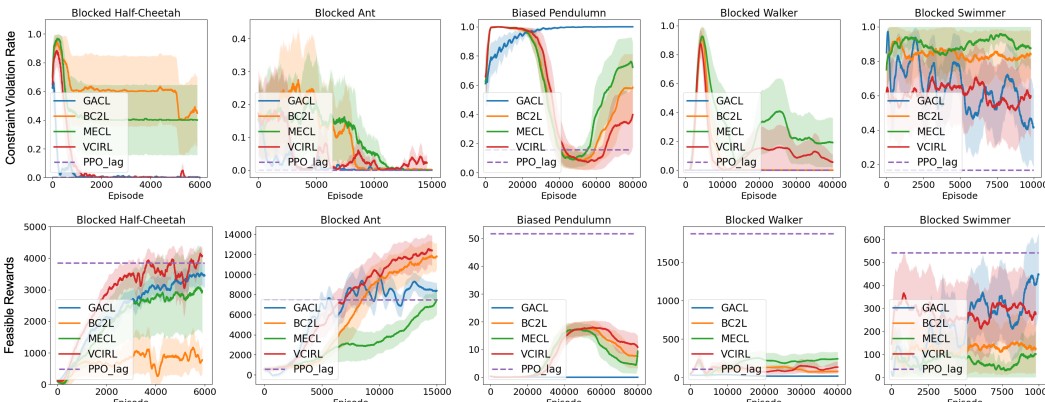

Figure 2: The constraint violation rate (top) and feasible rewards (i.e., the rewards from the trajectories without constraint violation, bottom) during training. From left to right, the environments are Blocked Half-cheetah, Blocked Ant, Bias Pendulum, Blocked Walker, and Blocked Swimmer.

Table 2: Testing performance. We report the average feasible rewards and the constraint violation rate in 100 runs. Check Appendix D.3 for the complete mean±std results. $\uparrow$ ($\downarrow$) indicates that a score is statistically greater (smaller) than the score achieved by VICRL with p-value $\leq$ 0.05 according to the Wilcoxon signed-rank test. (Table D.1 reports the p values.).

| | | Blocked Half-Cheetah | Blocked Ant | Biased Pendulum | Blocked Walker | Blocked Swimmer | HighD Speed | HighD Distance |
|---|---|---|---|---|---|---|---|---|
| Feasible Rewards | GACL | 3.48E+3$\downarrow$ | 7.21E+3 $\downarrow$ | 8.50E-1$\downarrow$ | 2.84E+1 $\downarrow$ | **5.78E+2**$\uparrow$ | -1.93E+1 $\downarrow$ | -1.70E+1 $\downarrow$ |
| | BC2L | 8.70E+2 $\downarrow$ | 1.20E+4 $\downarrow$ | 5.73E+0$\downarrow$ | 4.87E+1 $\downarrow$ | 1.41E+2$\downarrow$ | -2.93E-1 | 3.84E+0 $\downarrow$ |
| | MECL | 3.02E+3 $\downarrow$ | 8.55E+3 $\downarrow$ | 1.02E+0$\downarrow$ | **1.27E+2** $\uparrow$ | 6.37E+1$\downarrow$ | **9.67E-1** | 2.15E+0 $\downarrow$ |
| | VICRL | **3.81E+3** | **1.37E+4** | **6.64E+0** | 9.34E+1 | 1.91E+2 | -8.99E-1 | **4.60E+0** |
| Constraint Violation Rate | GACL | 0% | 0% | 100% $\uparrow$ | 0% | 42% $\downarrow$ | 14% | 19% $\downarrow$ |
| | BC2L | 47% $\uparrow$ | 0% | 58%$\uparrow$ | 0% $\downarrow$ | 84% $\uparrow$ | 33% $\uparrow$ | 33% |
| | MECL | 40% $\uparrow$ | 0% | 73% $\uparrow$ | 19% | 88% $\uparrow$ | 31% $\uparrow$ | 41% $\uparrow$ |
| | VICRL | 0% | 2% | 39% | 7% | 59% | 24% | 31% |

## 4.2 CONSTRAINT RECOVERY IN THE VIRTUAL ENVIRONMENT

Figure 2 and Table 2 show the training curves and the corresponding testing performance in each virtual environment. Compared to other baseline models, we find VICRL generally performs better with lower constraint violation rates and larger cumulative rewards. This is because VICRL captures the uncertainty of constraints by modeling their distributions and requiring the agent to satisfy all the sampled constraints, which facilitates a conservative imitation policy. Although MECL and GACL outperform VICRL in the Blocked Walker and the Blocked Swimmer environments, respectively, none of these algorithms can perform consistently better than the others. Figure D.5 visualizes the constraints learned by VICRL for a closer analysis.

### 4.3 CONSTRAINT RECOVERY FROM VIOLATING DEMONSTRATIONS

We use our virtual environment to study *"How well do the algorithms perform when the expert demonstrations may violate the true underlying constraint?"* Under the definition of ICRL problems, violation indicates that expert trajectories contain state-action pairs that do not satisfy the ground-truth constraint. The existence of violating expert trajectories is a crucial challenge for ICRL since in practice the expert data is noisy and there is no guarantee that all trajectories strictly follow underlying constraints. Our benchmark provides a testbed to study how the scale of violation influences the performance of ICRL baselines. To achieve this, we perform random actions during expert data generation so that the generated expert trajectories contain infeasible state-action pairs that violate ground-truth constraints.

Figure 3 shows the performance including the constraint violation rate (top row) and the feasible rewards (bottom row). We find the constraint violation rate (top row) increases significantly and the feasible rewards decrease as the scale of violation increases in the expert dataset, especially for GACL and BC2L, whose performance is particularly vulnerable to violating trajectories. Among the studied baselines, MECL is the most robust to expert violation, although its performance drops significantly when the violation rate reaches 80%. How to design an ICLR algorithm that is robust to expert violation remains a challenge for future work.

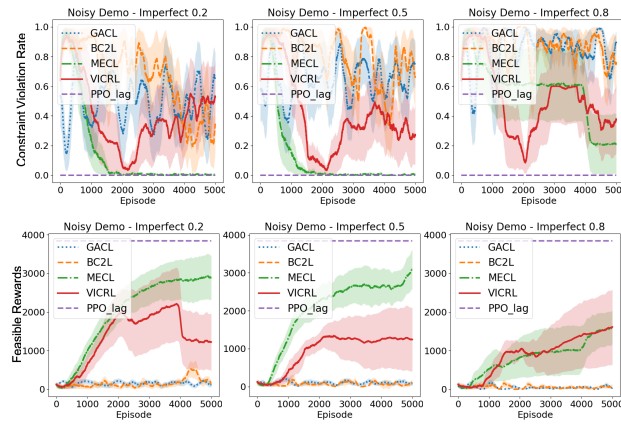

Figure 3: Model performance in the Blocked Half-Cheetah environment. From left to right, the percentages of trajectories containing violating state-action pairs are 20%, 50%, and 80%. Check Figure D.6 in Appendix for all results.

### 4.4 CONSTRAINT RECOVERY FROM STOCHASTIC ENVIRONMENTS

Our virtual environment can help answer the question *"How well do ICRL algorithms perform in stochastic environments?"* To achieve this, we modify the MuJoCo environments by adding noise to the transition functions at each step such that $p(s_{t+1}|s_t, a_t) = f(s_t, a_t) + \eta$, where $\eta \sim \mathcal{N}(\mu, \sigma)$. Under this design, our benchmark enables studying how the scale of stochasticity influences model performance by controlling the level of added noise. Figure 4 shows the results. We find ICRL models are generally robust to additive Gaussian noises in environment dynamics until they reach a threshold (e.g., $\mathcal{N}(0, 0.1)$). Another intriguing finding is that the constraint inference methods (MECL and B2CL) can benefit from a proper scale of random noise since these noisy signals induce stricter constraint functions and thus a lower constraint violation rate.

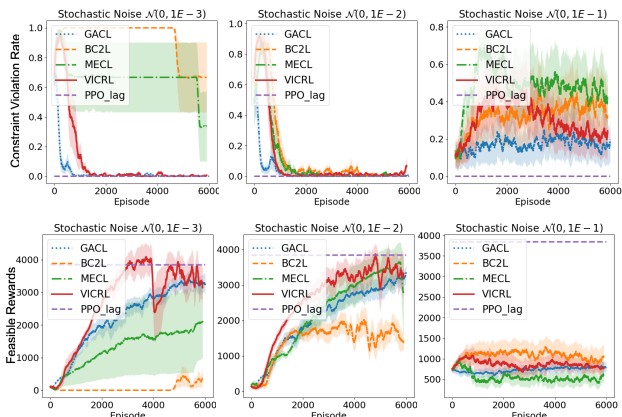

Figure 4: Model performance in the Blocked Half-Cheetah environment. From left to right, the transition function has the noises $\mathcal{N}(0, 0.001), \mathcal{N}(0, 0.01)$, and $\mathcal{N}(0, 0.1)$. Check Figure D.7 in Appendix for results in all environments.

## 5 REALISTIC ENVIRONMENT

Our realistic environment defines a highway driving task. This HighD environment examines if the agent can drive safely the ego car to the destination by following the constraints learned from human

drivers' trajectories (see Figure 5). In practice, many of these constraints are based on driving context and human experience. For example, human drivers tend to keep larger distances from trucks and drive slower on crowded roads. Adding these constraints to an auto-driving system can facilitate a more natural policy that resembles human preferences.

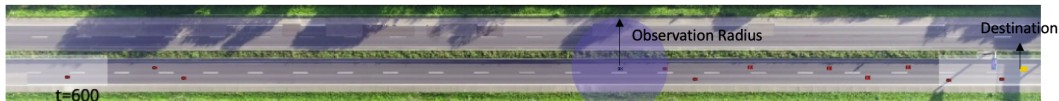

Figure 5: The Highway Driving (HighD) environment. The ego car is in blue, other cars are in red. The ego car can only observe the things within the region around (marked by blue). The goal is to drive the ego car to the destination (in yellow) without going off-road, colliding with other cars, or violating time limits and other constraints (e.g., speed and distance to other vehicles).

Table 3: The constraints for realistic environments.

| Type | Name | Dynamics | Obs. Dim. | Act. Dim. | Constraints |
|---|---|---|---|---|---|
| Realistic | HighD Velocity Constraint | Stochastic | 76 | 2 | Car Velocity $\leq 40$ m/s |
| | HighD Distance Constraint | Stochastic | 76 | 2 | Car Distance $\geq 20$ m |

**Environment Settings.** This environment is constructed by utilizing the HighD dataset (Krajewski et al., 2018). Within each recording, HighD contains information about the static background (e.g., the shape and the length of highways), the vehicles, and their trajectories. We break these recordings into 3,041 scenarios so that each scenario contains less than 1,000 time steps. To create the RL environment, we randomly select a scenario and an ego car for control in this scenario. The game context, which is constructed by following the background and the trajectories of other vehicles, reflects the driving environment in real life. To further imitate what autonomous vehicles can observe on the open road, we ensure the observed features in our environment are commonly used for autonomous driving (e.g., Speed and distances to nearby vehicles). These features reflect only partial information about the game context. To collect these features, we utilize the features collector from Commonroad RL (Wang et al., 2021). In this HighD environment, we mainly study a car Speed constraint and a car distance constraint (see Table 3) to ensure the ego car can drive at a safe speed and keep a proper distance from other vehicles. Section 5.2 further studies an environment having both of these constraints.

Note that the HighD environment is stochastic since 1) Human drivers might behave differently under the same context depending on the road conditions and their driving preferences. The population of drivers induces underlying transition dynamics that are stochastic. The trajectories in the HighD dataset are essentially samples from these stochastic transition dynamics. 2) Each time an environment is reset (either the game ends or the step limit is reached), it randomly picks a scenario with a set of driving trajectories. This is equivalent to sampling from the aforementioned transition dynamics.

**The significance of Constraints.** We show the difference in performance between a PPO-Lag agent (Section 3.1) that *knows* the ground-truth constraints and a PPO agent *without knowing* the constraints. Figure 6 reports the violation rate of the speed constraint (top left) and the distance constraint (top right). The bottom graphs report the cumulative rewards in both settings. We find 1) the PPO agent tends to violate the constraints in order to get more rewards and 2) the PPO-Lag agent abandons some of these rewards in order to satisfy the constraints. Their gap demonstrates the significance of these constraints. Appendix C.6 explains why these constraints are ideal by comparing them with other candidate constraint thresholds.

## 5.1 CONSTRAINT RECOVERY IN THE REALISTIC ENVIRONMENT

Figure 7 shows the training curves and Table 2 shows the testing performance. Among the studied methods, VICRL achieves a low constraint violation rate with a satisfying number of rewards. Although GACL has the lowest violation rate, it is at the cost of significantly degrading the controlling performance, which demonstrates that directly augmenting rewards with penalties (induced by constraints) can yield a control policy with much lower value. Appendix D.4 illustrates the causes of failures by showing the collision rate, time-out rate, and off-road rate. To illustrate how well the constraint is captured by the experimented algorithms, our plots include the upper bound of rewards

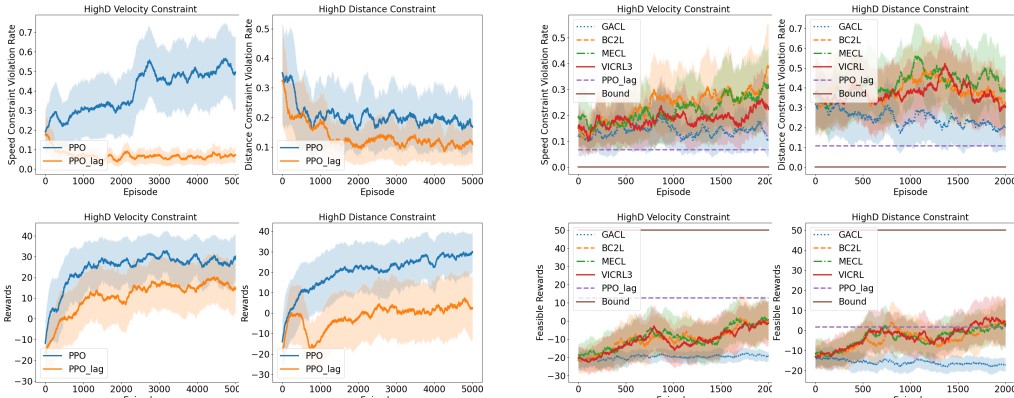

Figure 6: Model performance in the HighD environment with the speed (*left*) and distance (*right*) constraint.

Figure 7: The constraint violation rate (top) and feasible rewards (bottom) with the speed (left) and distance (right) constraints.

and the performance of the PPO-Lag agent (trained under the true constraints). It shows that there is sufficient space for future improvement under our benchmark.

## 5.2 Multiple Constraints Recovery

We consider the research question *"How well do ICRL algorithms work in terms of recovering multiple constraints?"*. Unlike the previously studied environments that include only one constraint, we extend the HighD environment to include both the speed and the distance constraints. To achieve this, we generate an expert dataset with the agent that considers both constraints by following 3.1 and test ICRL algorithms by using this dataset.

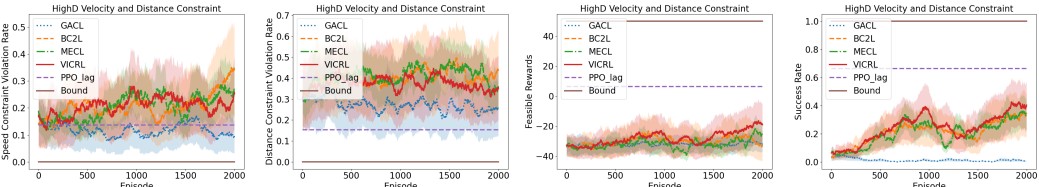

Figure 8: Model Performance in an environment with the speed and distance constraints. From left to right, we report speed and distance constraint violation rates, feasible rewards, and success rates.

Figure 8 shows the results. Compared to the performance of its single-constraint counterparts (in Figure 7), the rewards collected by the imitation policy are reduced significantly, although the constraint violation rate remains uninfluenced.

## 6 Conclusion

In this work, we introduced a benchmark, including robot control environments and highway driving environments, for evaluating ICRL algorithms. Each environment is aligned with a demonstration dataset generated by expert agents. To extend the Bayesian approach to constraint inference, we proposed VICRL to learn a distribution of constraints. The empirical evaluation showed the performance of ICRL algorithms under our benchmark.

## Acknowledgements

Resources used in preparing this research at the University of Waterloo were provided by Huawei Canada, the province of Ontario and the government of Canada through CIFAR and companies sponsoring the Vector Institute. Guiliang Liu's research was in part supported by the Start-up Fund UDF01002911 of the Chinese University of Hong Kong, Shenzhen. We would like to thank Guanren Qiao for providing valuable feedback for the experiments.

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

## A    RELATED WORK

In this section, we introduce the previous works that are most related to our research.

**Inferring Constraints from Demonstrations.** Previous works commonly inferred constraints to identify whether an action is allowed or a state is safe. Among these works, (Chou et al., 2018; Scobee & Sastry, 2020; McPherson et al., 2021; Park et al., 2019) are based on the *discrete* state-action space and constructed constraint sets to distinguish feasible state-action pairs from infeasible ones. Regarding *continuous* domains, the goal is to infer the boundaries between feasible and infeasible state-action pairs: (Lin et al., 2015; 2017; Armesto et al., 2017) estimated a constraint matrix from observations based on the projection of its null-space matrix. (Pérez-D'Arpino & Shah, 2017) learned geometric constraints by constructing a knowledge base from demonstration. (Menner et al., 2021) proposed to construct constraint sets that correspond to the convex hull of all observed data. (Malik et al., 2021; Gaurav et al., 2022) approximated constraints by learning neural functions from demonstrations. Some previous works (Calinon & Billard, 2008; Ye & Alterovitz, 2011; Pais et al., 2013; Li & Berenson, 2016; Mehr et al., 2016) focused on learning local trajectory-based constraints from a single trajectory. These works focus on inferring a *single candidate* constraint while some recent works learn a *distribution* over constraints, for example, (Glazier et al., 2021) learned a constraint distribution by assuming the environment constraint follows a logistic distribution. (Chou et al., 2020; Papadimitriou et al., 2021) utilized a Bayesian approach to update their belief over constraints, but these methods are restricted to discrete state spaces or toy environments like grid-worlds.

**Testing Environments for ICRL.** To the best of our knowledge, there is no common benchmark for ICRL, and thus previous works often define their own environments for evaluation, including 1) *Grid-Worlds* are the most popular environments due to their simplicity and interpretability. Previous works (Scobee & Sastry, 2020; McPherson et al., 2021; Papadimitriou et al., 2021; Glazier et al., 2021; Gaurav et al., 2022) added some obstacles to a grid map and examined whether their algorithms can locate these obstacles by observing expert demonstrations. However, it is difficult to generalize the model performance in these grid worlds to real applications with high-dimensional and continuous state spaces. 2) *Robotic Applications* have been used as test beds for constraint inference, for example, the manipulation of robot arms (Park et al., 2019; Menner et al., 2021; Armesto et al., 2017; Pérez-D'Arpino & Shah, 2017), quadrotors (Chou et al., 2019; 2020), and humanoid robot hands (Lin et al., 2017). However, there is no consistent type of robot for comparison, and the corresponding equipment is not commonly available. A recent work (Malik et al., 2021) used a *robotic simulator* by adding some pre-defined constraints into the simulated environments. Our virtual environments use a similar setting, but we cover more control tasks and include a detailed study of the environments and the added constraints. 3) *Safety-Gym (Ray et al., 2019)* is one of the most similar benchmarks to our work. However, Safety Gym is designed for validating *forward* policy-updating algorithms given some constraints, whereas our benchmark is designed for the *inverse* constraint-inference problem.

## B    DISCRETE ENVIRONMENTS

Our benchmark includes a Grid-World environment, which has a discrete state and action space. Although migrating the model performance to real-world applications is difficult, Grid-Worlds are commonly studied RL environments where we can visualize the recovered constraints and the trajectories generated by agents. Our benchmark uses a Grid-World to answer the question *"How well do the ICRL algorithms perform in terms of recovering the exact least constraining constraint?"*

### B.1    ENVIRONMENT SETTINGS

Our benchmark constructs a map of size $7 * 7$ and four different constraint maps (top row Figure B.1) for testing the baseline methods. For benchmarking ICRL algorithms, each environment is accompanied by a demonstration dataset of expert trajectories generated with the PI-Lag algorithm 2 (see Section 3.1). Note that to be compatible with previous work that studied Grid-World environments Scobee & Sastry (2020), we replace the policy gradient algorithm in the baseline algorithms with policy iteration for solving discretized control problems.

### B.2    EXPERIMENT RESULTS

To study how well the ICLR algorithms perform in terms of recovering the exact least constraining constraint, we visualize the ground truth constraint map and the constraint maps recovered by ICLR baselines in Figure B.1. (For other metrics, please find the constraint violation rate and feasible cumulative rewards in Figure D.3, and the generated trajectories in Figure D.2.). We find the difference between the added constraint (top row Figure B.1) and the recovered constraint is significant, although most algorithms (BC2L, MECL, and VICRL) learn a policy that matches well the policy of an expert agent. In most settings, the size of the recovered constraint set is larger than the ground-truth constraint (i.e., constraint learning is too conservative). While baselines including MECL and VICRL integrated regularization about the size of the constraint set into their loss, the results show that the impact of this regularization is limited, and there is plenty of room for improvement.

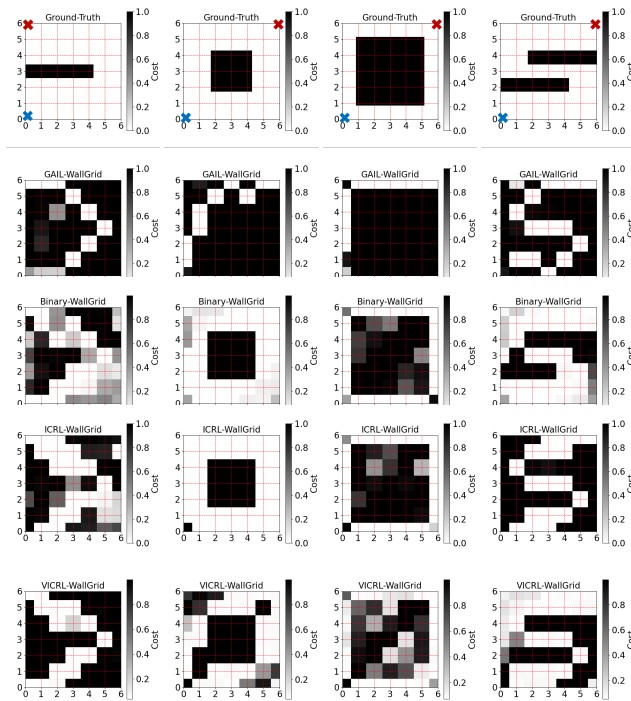

Figure B.1: The recovered constraints under 4 settings (from the left to right columns). From the second to the last row, the experimented methods are GACL, BC2L, MECL, and VICRL. Blue and red mark the starting and target locations.

## C  MORE IMPLEMENTATION AND ENVIRONMENT DETAILS

### C.1  MORE INFORMATION ABOUT THE VIRTUAL ENVIRONMENTS

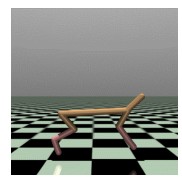 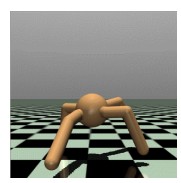 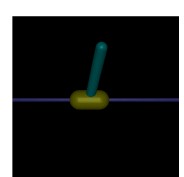 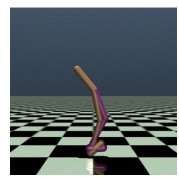 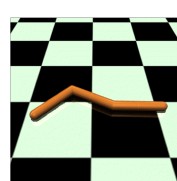

Figure C.1: Mujoco environments. From left to right, the environments are Half-cheetah, Ant, Inverted Pendulum, Walker and Swimmer.

Our virtual environments are based on Mujoco (see Figure C.1). We provide more details about the virtual environments as follows:

- *Blocked Half-Cheetah.* The agent controls a robot with two legs. The reward is determined by the distance it walks between the current and the previous time step and a penalty over the magnitude of the input action. The game ends when a maximum time step (1000) is reached. We define a constraint that blocks the region with X-coordinate $\leq -3$, so the robot is only allowed to move in the region with X-coordinate between -3 and $\infty$.

- *Blocked Ant.* The agent controls a robot with four legs. The rewards are determined by the distance to the origin and a healthy bonus that encourages the robot to stay balanced. The game ends when a maximum time step (500) is reached. Similar to the Blocked Half-Cheetah environment, we define a constraint that blocks the region with X-coordinate $\leq -3$, so the robot is only allowed to move in the region with X-coordinate between -3 and $\infty$.

- *Biased Pendulum.* Similar to the Gym CartPole (Brockman et al., 2016), the agent's goal is to balance a pole on a cart. The game ends when the pole falls or a maximum time step (100) is reached. At each step, the environment provides a reward of 0.1 if the X-coordinate $\geq 0$ and a reward of 1 if the X-coordinate $\leq -0.01$. The reward monotonically increases from 0.1 to 1 when $-0.01 <$ X-coordinate $< 0$. We define a constraint that blocks the region with X-coordinate $\leq -0.015$, so the reward incentivizes the cart to move left, but the constraint prevents it from moving too far. If the agent can detect the ground-truth constraint threshold, it will drive the cart to move into the region with X-coordinate between $-0.015$ and $-0.01$ and stay balanced there.

- *Blocked Walker.* The agent controls a robot with two legs and learns how to make the robot walk. The reward is determined by the distance it walks between the current and the previous time step and a penalty over the magnitude of the input action (this is following the original Walker2d environment). The game ends when the robot loses its balance or reaches a maximum time step (500). Similar to the Blocked Half-Cheetah and Blocked Ant environment, we constrain the region with X-coordinate $\leq -3$, so the robot is only allowed to move in the region with X-coordinate between -3 and $\infty$.

- *Blocked Swimmer.* The agent controls a robot with two rotors (connecting three segments) and learns how to move. The reward is determined by the distance it walks between the current and the previous time step and a penalty over the magnitude of the input action. The game ends when the robot reaches a maximum time step (500). Unlike the Blocked Half-Cheetah and Blocked Ant environment, it is easier for the Swimmer robot to move ahead than move back, and thus we constrain the region with X-coordinate $\geq 0.5$, so the robot is only allowed to move in the region with X-coordinate between $-\infty$ and $0.5$.

## C.2 MORE ALGORITHM

We show the PI-Lag in Algorithm 2.

---

**Algorithm 1:** Proximal Policy Optimization Lagrange (PPO-Lag)

**Input:** Constraint function $f^*$, constraint threshold $\epsilon$, Lagrange multiplier $\lambda$, rollout rounds $B$, update rounds $\mathcal{K}$, loss parameters $\xi_1$ and $\xi_2$, clipping parameter $\omega$, imitation policy $\pi_\theta$, value functions $V_r$ and $V_c$;

Initialize state $s_0$ from CMDP and the roll-out dataset $\mathcal{D}_{roll}$;

**for** $b = 1, 2, \ldots, B$ **do**

    Perform Monte-Carlo roll-out with the policy $\pi_\theta$ in the environment;

    Collect trajectories $\tau_b = [s_0, a_0, r_0, c_0, \ldots, s_T, a_T, r_T, c_T]$ where $c_t = f^*(s_t, a_t)$;

    Calculate reward advantages $A_t^r$, total rewards $R_t$, constraint advantages $A_t^c$ and total costs $C_t$ from the trajectory;

    Add samples to the dataset $\mathcal{D}_{roll} = \mathcal{D}_{roll} \cup \{s_t, a_t, r_t, A_t^r, R_t, c_t, A_t^c, C_t\}_{t=1}^T$;

**end**

**for** $\kappa = 1, 2, \ldots, \mathcal{K}$ **do**

    Sample a data point $s_\kappa, a_\kappa, r_\kappa, A_\kappa^r, R_\kappa, c_\kappa, A_\kappa^c, C_\kappa$ from the dataset $\mathcal{D}_{roll}$;

    Calculate the clipping loss

    $L^{CLIP} = \min\left[\frac{\pi(a_\kappa|s_\kappa)}{\pi_{old}(a_\kappa|s_\kappa)}(\hat{A}_\kappa^r + \lambda\hat{A}_\kappa^c), clip(\frac{\pi(a_\kappa|s_\kappa)}{\pi_{old}(a_\kappa|s_\kappa)}, 1-\omega, 1+\omega)(\hat{A}_\kappa^r + \lambda\hat{A}_\kappa^c)\right]$;

    Calculate the value function loss $L^{VF} = \|V_\theta^r(s_\kappa) - R_\kappa\|_2^2 + \|V_\theta^c(s_\kappa) - C_\kappa\|_2^2$;

    Update policy parameters $\theta$ by minimizing the loss: $-L^{CLIP} + \xi_1 L^{VF} - \xi_2 \mathcal{H}(\pi)$;

**end**

Update the Lagrange multiplier $\lambda$ by minimizing the loss $L^\lambda$: $\lambda[\mathbb{E}_{\mathcal{D}_{roll}}(\hat{A}^c) - \epsilon]$;

---

## C.3 HYPER-PARAMETERS

We published our benchmarks, including the configurations of the environments and the models at `https://github.com/Guiliang/ICRL-benchmarks-public`.Please see the README.MD file for more details. We provide a brief summary of the hyper-parameters.

---

**Algorithm 2:** Policy Iteration Lagrange(PI-Lag)

---

**Input:** Constraint function $f^*$, Lagrange multiplier $\lambda$ rollout rounds $B$, update rounds $\mathcal{K}$, loss
      parameters $\xi_1$ and $\xi_2$, imitation policy $\pi_\theta$;

Initialize state $s_0$ from CMDP and the roll-out dataset $\mathcal{D}_{roll}$;

Initialize Values $V(s) \in \mathbf{R}$ and $\pi(s) \in \mathcal{A}(s)$ for all $s \in \mathcal{S}$;

**while** *not converge;*                       `// Policy evaluation.`
 **do**
    **for** $s \in S$ **do**
        $V(s) = \sum_{r,s'} p(s', r|s, \pi(s))[r - \lambda c^* + \gamma V(s')]$ where $c_t^* = f^*(s_t)$;
    **end**
**end**

**while** *not converge;*                       `// Policy update.`
 **do**
    **for** $s \in S$ **do**
        $\pi(s) = \arg\max_a \sum_{r,s'} p(s', r|s, \pi(s))[r - \lambda c^* + \gamma V(s')]$ where $c_t^* = f^*(s_t)$;
    **end**
**end**

**while** *not converge;*                   `// Lagrange multiplier update.`
 **do**
    **for** $b = 1, 2, \ldots, B$ **do**
        Collect trajectories $\tau_b = [s_0, a_0, c_0^*, \ldots, s_T, a_T, c_T^*]$ where $c_t^* = f^*(s_t)$;
        Calculate total costs $C_t$ from the trajectory from $\tau_b$;
        Add samples to the dataset $\mathcal{D}_{roll} = \mathcal{D}_{roll} \cup \{s_t, a_t, C_t\}_{t=1}^T$;
    **end**
    Update the Lagrange multiplier $\lambda$ by minimizing the loss $L^\lambda$: $\lambda[\mathbb{E}_{\mathcal{D}_{roll}}(C_t) - \epsilon]$;
**end**

---

In order to develop a fair comparison among ICRL algorithms, we use the same setting for all algorithms.

In the **virtual environments**, we set 1) the batch size of PPO-Lag to 64, 2) the size of the hidden layer to 64, and 3) the number of hidden layers for the policy function, the value function, and the cost function to 3. We decide the other parameters, including the learning rate of both PPO-Lag and constraint model, by following some previous work (Malik et al., 2021) and their implementation. The random seeds of virtual environments are 123, 321, 456, 654, and 666.

In the **realistic environments**, we set 1) the batch size of the constraint model to 1000, 2) the size of the hidden layer to 64 and 3) the number of hidden layers for the policy function, the value function and the cost function to 3. We decide the other parameters, including the learning rate of both PPO-Lag and constraint model, by following CommonRoad RL (Wang et al., 2021) and their implementation. During our experiment, we received plenty of help from their forum [2]. We will acknowledge their help in the formal version of this paper. The random seeds of realistic environments are 123, 321, and 666.

## C.4   Experimental Equipment and Infrastructures

We run the experiment on a cluster operated by the Slurm workload manager. The cluster has multiple kinds of GPUs, including Tesla T4 with 16 GB memory, Tesla P100 with 12 GB memory, and RTX 6000 with 24 GB memory. We used machines with 12 GB of memory for training the ICRL models. The number of running nodes is 1, and the number of CPUs requested per task is 16. Given the aforementioned resources, running one seed in the virtual environments and the realistic environments takes 2-4 hours and 10-12 hours respectively.

---

[2] *https://gitlab.lrz.de/tum-cps/commonroad-rl*

### C.5 COMPUTATIONAL COMPLEXITY

We provide a brief analysis of the computational complexity. The ICRL algorithms, including GACL, MECL, BC2L, and VICRL, use an iterative updating paradigm and thus their computational complexities are similar. Let $K$ denote the number of iterations. Within each iteration, the algorithms update both the imitation policy and the constraint model. Let $M$ denote the number of episodes that the PPO-Lag algorithm runs in the environments. Let $N$ denote the number of sampling and expert trajectories. Let $L$ denote the maximum length of each trajectory. During training, we use mini-batch gradient descent. Let $B$ denote the batch size, and then the computational complexity is $O(KL(M+N)/B)$.

### C.6 EXPLORING OTHER CONSTRAINTS IN THE REALISTIC ENVIRONMENTS

The constraint thresholds in our environments are determined empirically according to the performance (constraint violation rate and rewards) of the PPO agent and the PPO-Lag agent. To support this claim, we show the performance of other thresholds and analyze why they are sub-optimal in terms of validating ICRL algorithms.

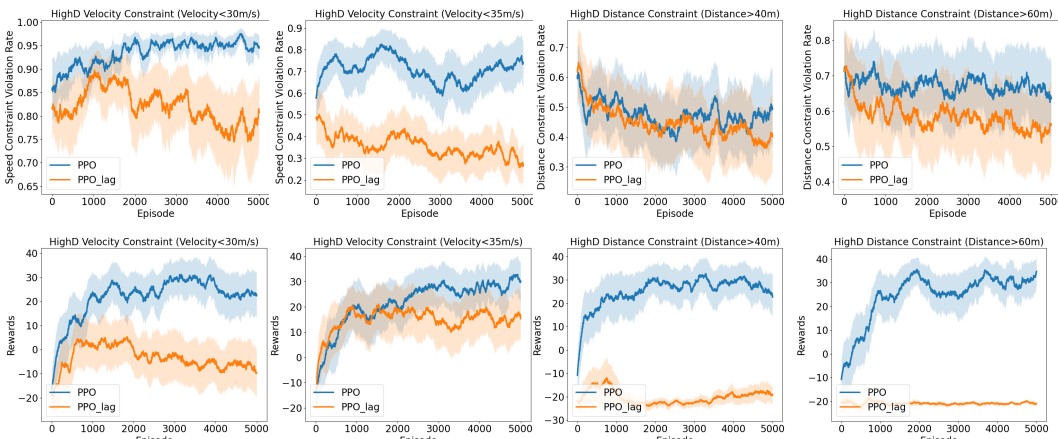

Figure C.2: From left to right, the constraint violation rate (top) and rewards (bottom) of the PPO and PPO-Lag agents in the HighD environments with constraints 1) Ego Car Velocity < 30 m/s, 2) Ego Car Velocity < 35 m/s, 3) Car Distance > 40 m, and 4) Car Distance > 60 m.

We have explored the option of using a 30m/s velocity constraint (The first column on the left in Figure C.2) and 35m/s velocity constraint (The second column on the left in Figure C.2). Ideally, these constraints should be closer to the realistic speed limit in most countries. However, the HighD dataset comes from German highways where there is no speed limit. Moreover, when building the environment, the ego car is accompanied by an initial speed calculated from the dataset. We observed that the initial speed is already higher than the speed limit (e.g., 35m/s) in many scenarios, and thus the violation rate will always be 1 in these scenarios, leaving no opportunity for improving the policy. This explains why the corresponding violation rates are high for the PPO and the PPO-Lag agents.

We also explored the option of using a 40m distance constraint (third column in Figure C.2) and a 60m distance constraint (fourth column in Figure C.2). Ideally, these constraints should be more consistent with the 2-second gap recommendation (the average speed is around 30m/s in HighD, so the recommended gap is 2*30m/s=60m), but we find the controlling performance of the PPO-Lag agents are very limited, which shows the agent cannot even develop a satisfying control policy when knowing the ground-truth constraints. This is because the ego car learns to frequently go off-road in order to maintain the large gap.

# D    MORE EXPERIMENTAL RESULTS

## D.1    ADDITIONAL EXPERIMENTAL RESULTS IN THE VIRTUAL ENVIRONMENTS

Figure D.1 shows the additional experimental results in the virtual environment.

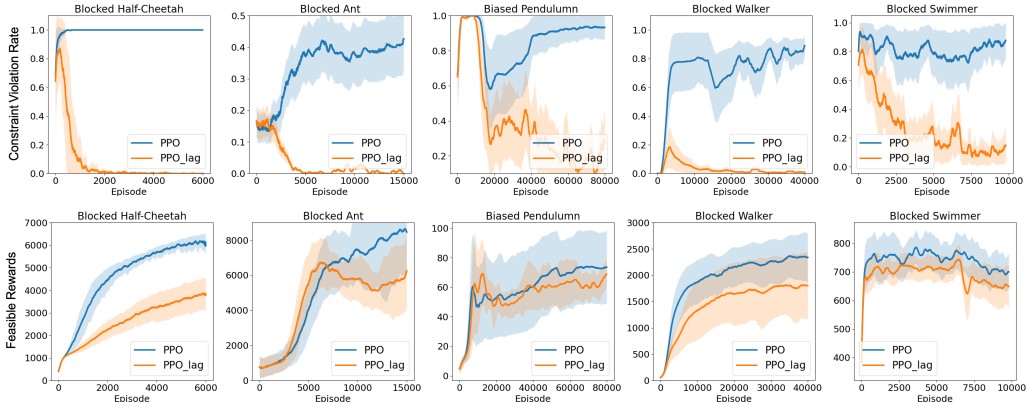

Figure D.1: The constraint violation rate (top) and rewards (bottom). Environments from left to right: Blocked Half-cheetah, Blocked Ant, Biased Pendulum, Blocked Walker, and Blocked Swimmer.

## D.2    ADDITIONAL EXPERIMENTAL RESULTS IN THE DISCRETE ENVIRONMENTS

Figure D.2 and Figure D.3 show the additional experimental results in the discrete environment.

## D.3    THE COMPLETE RESULTS FOR TESTING PERFORMANCE

Table D.2, Table D.2, Table D.3, Table D.4, and Table D.5 show the complete results for the testing performance.

Table D.1: The p Values of Wilcoxon signed-rank test across 100 runs. We repeat the test for the models trained by one random seed (we have a total of 5 random seeds) and report the averaged p values.

|  |  | Blocked Half-Cheetah | Blocked Ant | Biased Pendulum | Blocked Walker | Blocked Swimmer | HighD Velocity | HighD Distance |
|---|---|---|---|---|---|---|---|---|
| Feasible Rewards | GACL | 1.17E-2 | 1.23E-06 | 2.46E-14 | 3.90E-18 | 3.42E-4 | 8.33E-5 | 3.28E-05 |
|  | BC2L | 1.40E-15 | 2.98E-05 | 4.89E-4 | 2.34E-8 | 9.98E-10 | 5.75E-1 | 3.43E-2 |
|  | MECL | 4.07E-2 | 3.72E-08 | 2.46E-14 | 2.94E-2 | 7.27E-3 | 3.40E-1 | 4.86E-2 |
| Constraint Violation Rate | GACL | 1.57E-1 | 6.13E-2 | 1.71E-18 | 7.68E-2 | 1.48E-4 | 2.81E-1 | 1.53E-2 |
|  | BC2L | 7.93E-2 | 8.17E-1 | 1.84E-06 | 1.21E-7 | 8.79E-13 | 3.65E-2 | 5.54E-2 |
|  | MECL | 1.52E-23 | 5.20E-1 | 1.71E-18 | 9.98E-2 | 1.55E-4 | 1.72E-2 | 7.16E-3 |

Table D.2: Testing performance in the virtual environments. We report the feasible rewards (i.e., the rewards from the trajectories without constraint violation) computed with 50 runs.

|  | Half-cheetah | Blocked Ant | Biased Pendulum | Blocked Walker | Blocked Swimmer |
|---|---|---|---|---|---|
| GACL | $3477.53 \pm 416.54$ | $7213.62 \pm 993.12$ | $0.85 \pm 0.02$ | $28.35 \pm 0.77$ | $578.27 \pm 148.16$ |
| BC2L | $870.09 \pm 499.03$ | $11956.26 \pm 1980.88$ | $5.73 \pm 5.60$ | $48.73 \pm 4.18$ | $141.82 \pm 152.14$ |
| MECL | $3024.88 \pm 1364.59$ | $8546.19 \pm 1262.03$ | $1.02 \pm 1.63$ | $126.76 \pm 52.21$ | $63.66 \pm 107.95$ |
| VICRL | $3805.72 \pm 511.66$ | $13670.32 \pm 2511.89$ | $6.64 \pm 4.45$ | $93.40 \pm 93.97$ | $191.11 \pm 154.57$ |

## D.4    COMPLEMENTARY RESULTS IN THE REALISTIC ENVIRONMENT

Figure D.4 reports the average velocity, collision rate, off-road rate, time-out rate and goal-reaching rate during training. We find the off-road rate of GACL is significantly higher than other methods. It

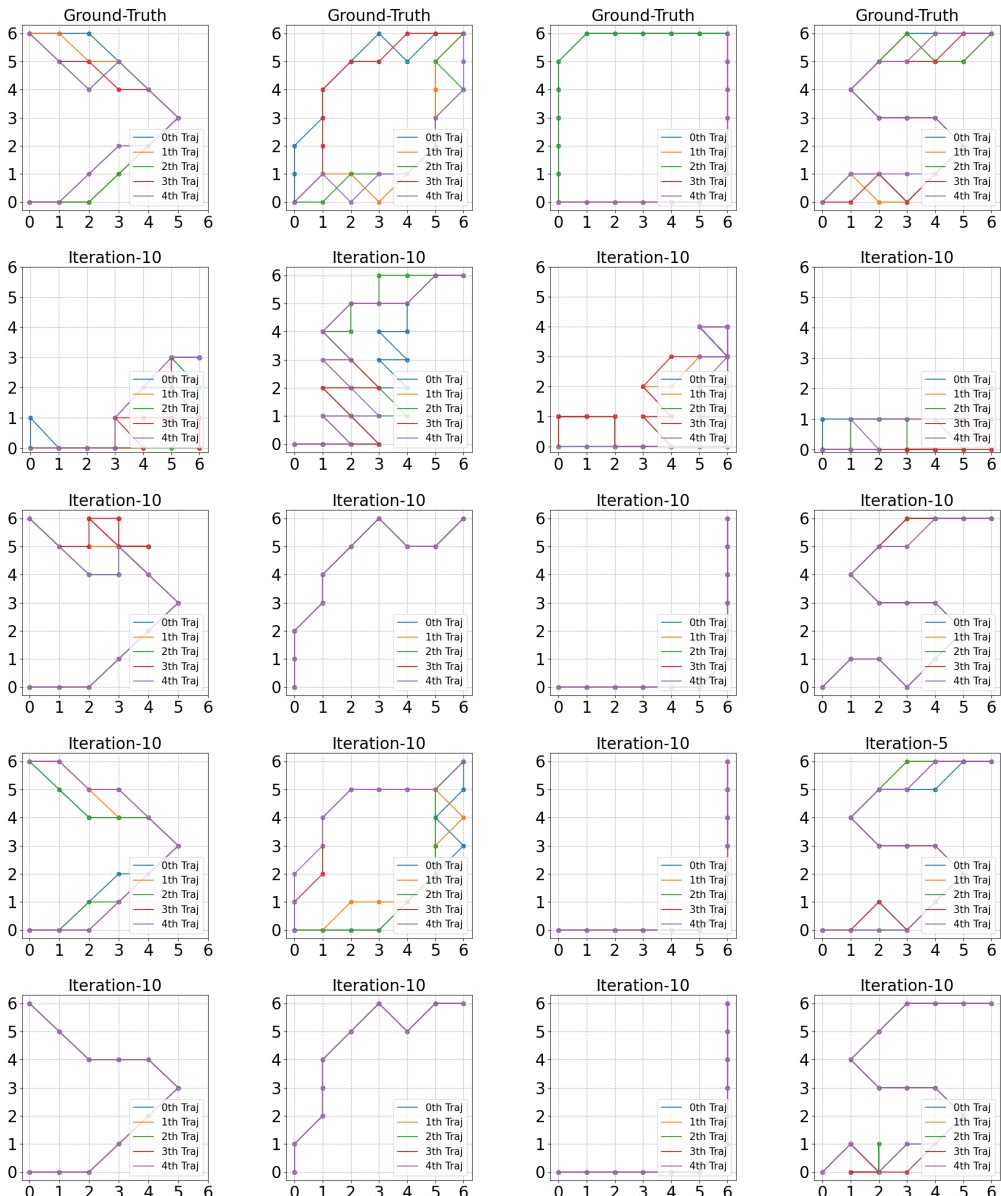

Figure D.2: The trajectories generated by different agents in the discrete environments.

Table D.3: Testing performance in the virtual environments. We report the constraint violation rate computed with 50 runs.

| | Half-cheetah | Blocked Ant | Biased Pendulum | Blocked Walker | Blocked Swimmer |
|---|---|---|---|---|---|
| GACL | $0.0 \pm 0.0$ | $0.0 \pm 0.0$ | $1.0 \pm 0.0$ | $0.0 \pm 0.0$ | $0.42 \pm 0.23$ |
| BC2L | $0.47 \pm 0.24$ | $0.0 \pm 0.0$ | $0.58 \pm 0.23$ | $0.0 \pm 0.0$ | $0.84 \pm 0.14$ |
| MECL | $0.40 \pm 0.24$ | $0.0 \pm 0.0$ | $0.73 \pm 0.17$ | $0.19 \pm 0.17$ | $0.88 \pm 0.12$ |
| VICRL | $0.0 \pm 0.0$ | $0.02 \pm 0.02$ | $0.39 \pm 0.22$ | $0.07 \pm 0.07$ | $0.59 \pm 0.23$ |

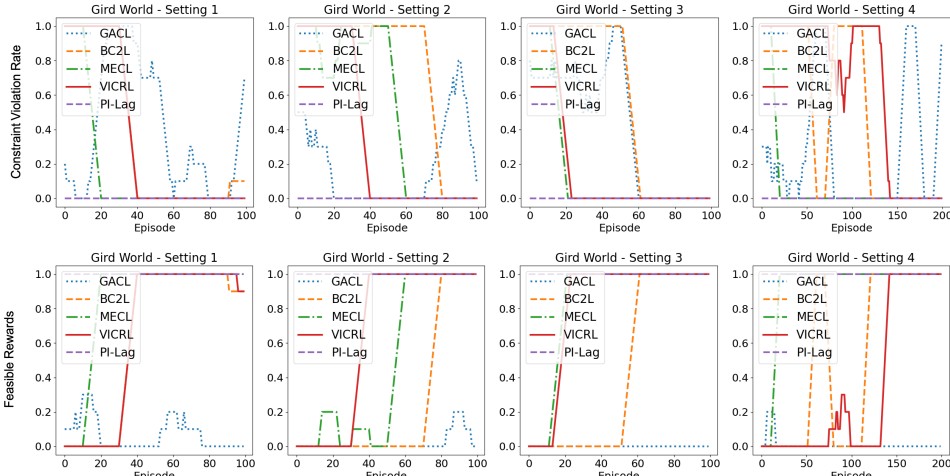

Figure D.3: The constraint violation rate (top) and feasible rewards (i.e., the rewards from the trajectories without constraint violation, bottom) during training.

Table D.4: Testing performance in the realistic environments. We report the feasible rewards (i.e., the rewards from the trajectories without constraint violation) computed with 50 runs.

|  | HighD Velocity Constraint | HighD Distance Constraint |
|---|---|---|
| GACL | -19.13 ± 2.99 | -17.02 ± 3.31 |
| BC2L | -0.29 ± 11.18 | 3.84 ± 11.28 |
| MECL | 0.97 ± 11.48 | 2.15 ± 10.45 |
| VICRL | -0.90 ± 11.80 | 4.60 ± 11.71 |

Table D.5: Testing performance in the realistic environments. We report the constraint violation rate computed with 50 runs.

|  | HighD Velocity Constraint | HighD Distance Constraint |
|---|---|---|
| GACL | 0.14 ± 0.09 | 0.19 ± 0.11 |
| BC2L | 0.33 ± 0.15 | 0.33 ± 0.15 |
| MECL | 0.31 ± 0.15 | 0.41 ± 0.17 |
| VICRL | 0.24 ± 0.12 | 0.31 ± 0.15 |

explains why GACL cannot achieve a satisfying performance. Another main limitation of current baselines is their incapability of preventing the collision events, especially under the car distance constraints.

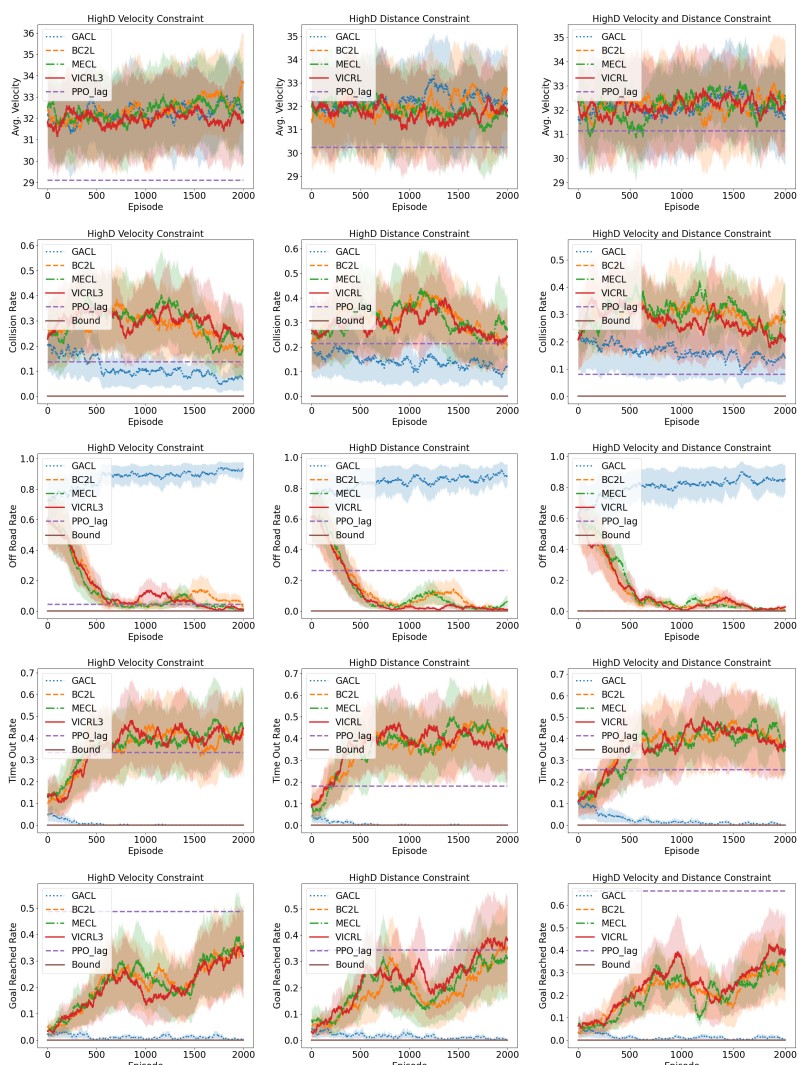

Figure D.4: The average velocity (first column), collision rate (second column), off road rate (third column), time out rate (fourth column) and goal reaching rate (last column) during training. From left to right, the environments are HighD with the velocity constraints, the distance constraints and both of these constraints.

## D.5    CONSTRAINT VISUALIZATION

Figure D.5 visualizes the learned constraints with 1) partial dependency plots (red curve, on the left) accompanied by the samples from the constraint distribution (blue points, i.e., $P_{\mathcal{C}}(c|s, a)$, marked by cost) and 2) histograms (blue, on the right) showing the number of states with a specific feature value (e.g., x position) during testing. The x-axis of these plots show the features where the ground-truth constraints are defined on (this message is hidden from the agents during training). In order to understand how well the constraints are captured, we can compare these plots with the definition of ground truth constraints in Table 1 and Table 3.

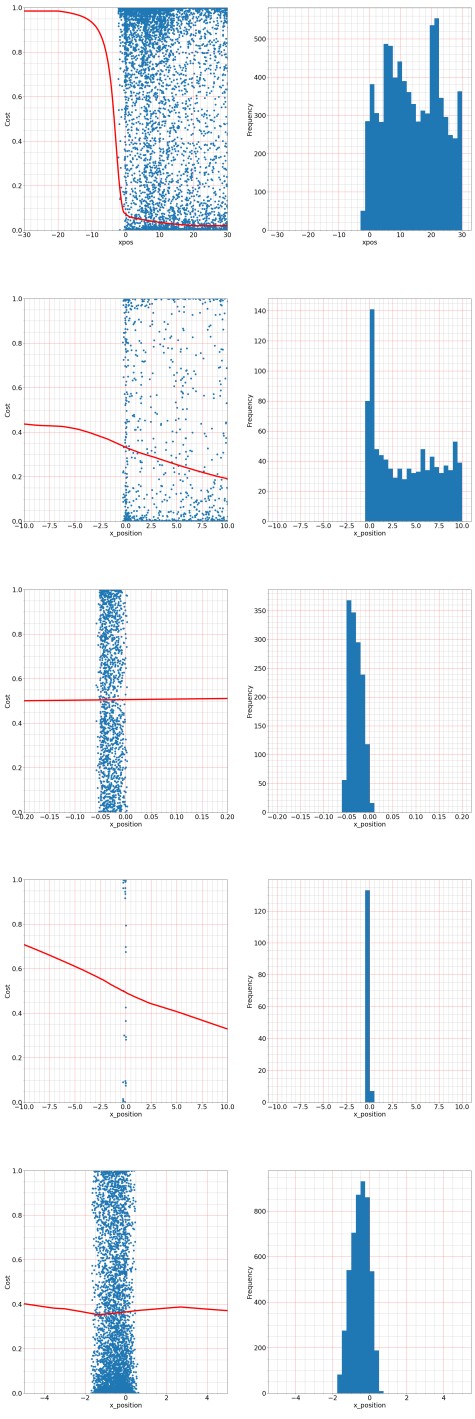

Figure D.5: Visualization the learned constraints by a pair of plots for VICRL-RS (left column) and (right column) VICRL-VaR. Each pair includes 1) partial dependency plots (red curve, on the left) accompanied by the samples from the constraint distribution (blue points, i.e., $P_{\mathcal{C}}(c|s,a)$, marked by cost) and 2) histograms (blue, on the right) showing the number of states with a specific feature value (e.g., x position) during testing. From top to bottom, the testing environments are Blocked Half-Cheetah, Blocked Ant, Biased Pendulum, Blocked Walker and Blocked Swimmer.

### D.6 Constraint Recovery From Violating Demonstrations In Other Four Environments

Figure D.6: The text is the result of the Half-Cheetah environment, and the following is the result of the other four environments. From top to bottom is Blocked Ant, Blocked Walker, Blocked Swimmer and Biased Pendulum.

### D.7 Constraint Recovery From Stochastic Environment In Other Four Environments

Figure D.7: The text is the result of the Half-Cheetah environment, and the following is the result of the other four environments. From top to bottom is Blocked Ant, Blocked Walker, Blocked Swimmer and Biased Pendulum respectively.

## E Limitations, Challenges and Open Questions

We introduce limitations and challenges in ICRL, as well as open questions for future work.

**Constraint Violation.** The imitation policies of ICRL agents are updated with RCPO (Tessler et al., 2019), but Lagrange relaxation methods are sensitive to the initialization of the Lagrange multipliers and the learning rate. There is no guarantee that the imitation policies can consistently satisfy the given constraints (Liu et al., 2021). As a result, even when a learned constraint function matches the ground-truth constraint, the learned policy may not match the expert policy, causing significant variation in training and sub-optimal model convergence. If we replace the Lagrange relaxation with Constrained Policy Optimization (CPO) (Achiam et al., 2017; Chow et al., 2019; Yang et al., 2020; Liu et al., 2020), ICRL may not finish training within a reasonable amount of time since CPO is computationally more expensive. How to design an efficient policy learning method that matches ICRL's iterative updating paradigm will be an important future direction.

**Unrealistic Assumptions about Expert Demonstrations.** ICRL algorithms typically assume that the expert demonstrations are optimal in terms of satisfying the constraints and maximizing rewards. There is no guarantee that these assumptions hold in practice since many expert agents (e.g., humans) do not always strive for optimality and constraint satisfaction. Previous works (Brown et al., 2019a;b; Wu et al., 2019; Chen et al., 2020; Tangkaratt et al., 2020; 2021), introduced IRL approaches to learn rewards from sub-optimal demonstrations, but how to extend these methods to constraint inference is unclear. A promising direction is to model *soft* constraints that assume that expert agents only follow the constraints with a certain probability.

**Insufficient Constraint Diversity.** ICRL can potentially recover complex constraints, but our benchmark mainly considers linear constraints as the ground-truth constraints (although this information is hidden from the agent). Despite this simplification, our benchmark is still very challenging: a ICRL agent must identify relevant features (e.g., velocity in x and y coordinates) among all input features (78 in total) and recover the exact constraint threshold (e.g., 40 m/s). For future work, we will explore nonlinear constraints and constraints on high-dimensional input spaces (e.g., pixels).

**Online versus Offline ICRL.** ICRL algorithms commonly learn an imitation policy by interacting with the environment. The online training nevertheless contradicts with the setting of many realistic applications where only the demonstration data instead of the environment is available. Given the recent progress in offline IRL (Jain et al., 2019; Lee et al., 2019; Kostrikov et al., 2020; Garg et al., 2021), extending ICRL to the offline training setting will be an important future direction.

## F Societal Impact

**Positive Societal Impacts** The ability to discover what can be done and what cannot be done is an important function of modern AI systems, especially for systems that have frequent interactions with humans (e.g., house keeping robots and smart home systems). As an important stepping stone towards the design of effective systems, constraint models can help develop human-friendly AI systems and facilitate their deployments in real applications.

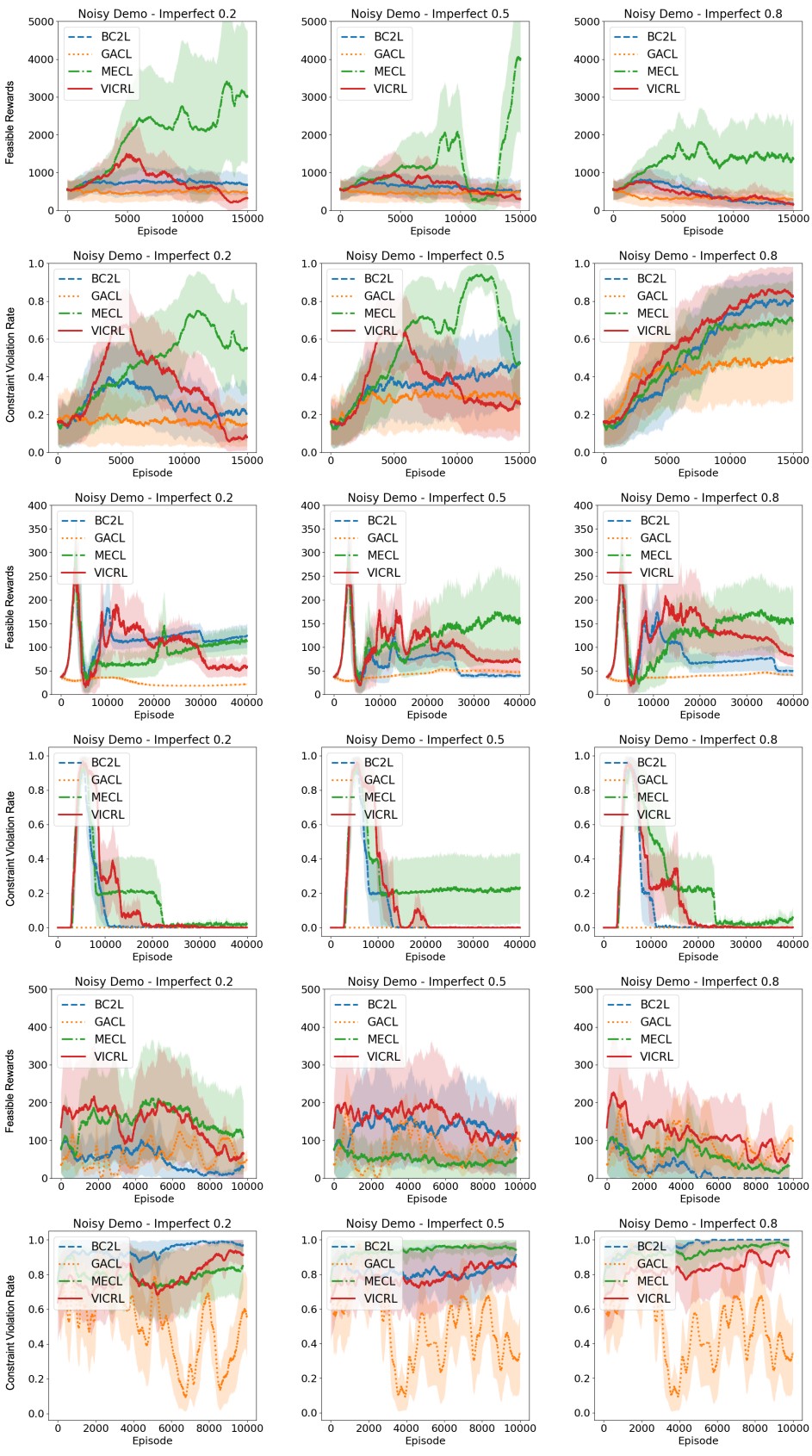

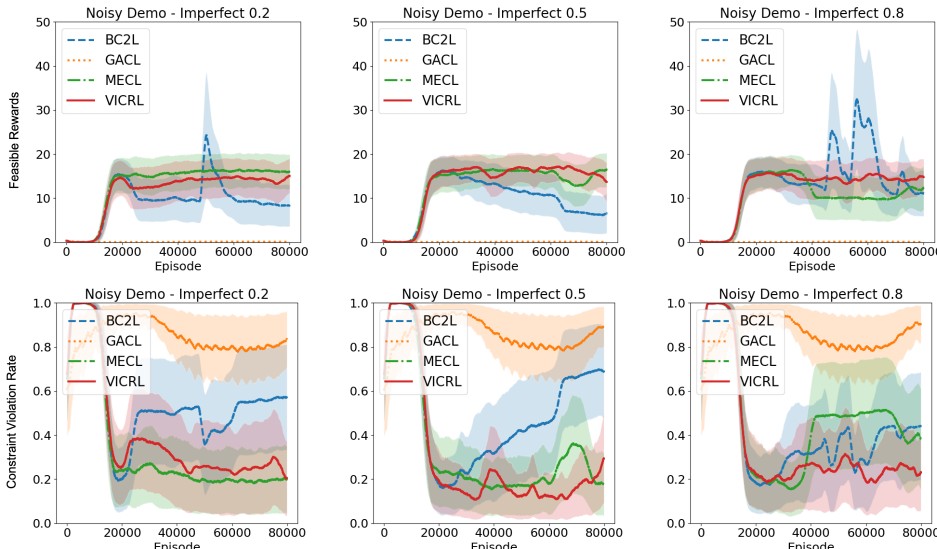

Figure D.6: From left to right, the percentages of trajectories containing violating state-action pairs are 20%, 50%, and 80%. The environment from top to bottom is Blocked Ant, Blocked Walker, Blocked Swimmer and Biased Pendulum. Feasible rewards(top) and constraint violation rate(bottom) are two metrics during training.

**Negative Societal Impacts** Possible real-world applications of constraint models include autonomous driving systems. Since constraint models are often represented by black-box deep models, there is no guarantee that the models are trustworthy and interpretable. When an autonomous vehicle is involved into an accident, it is difficult to identify the cause of this accident, which might cause a loss of confidence in autonomous systems while negatively impacting society.

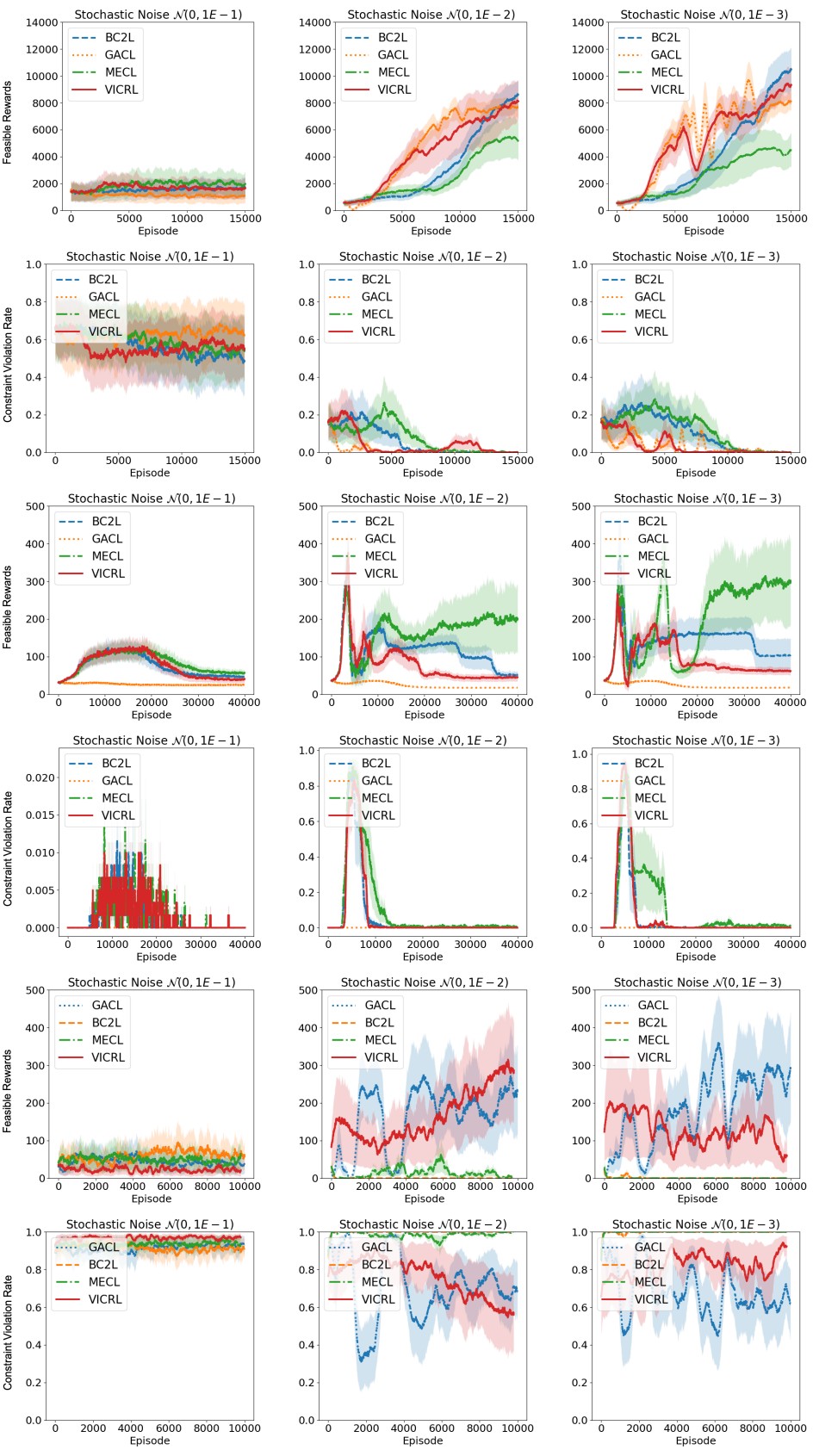

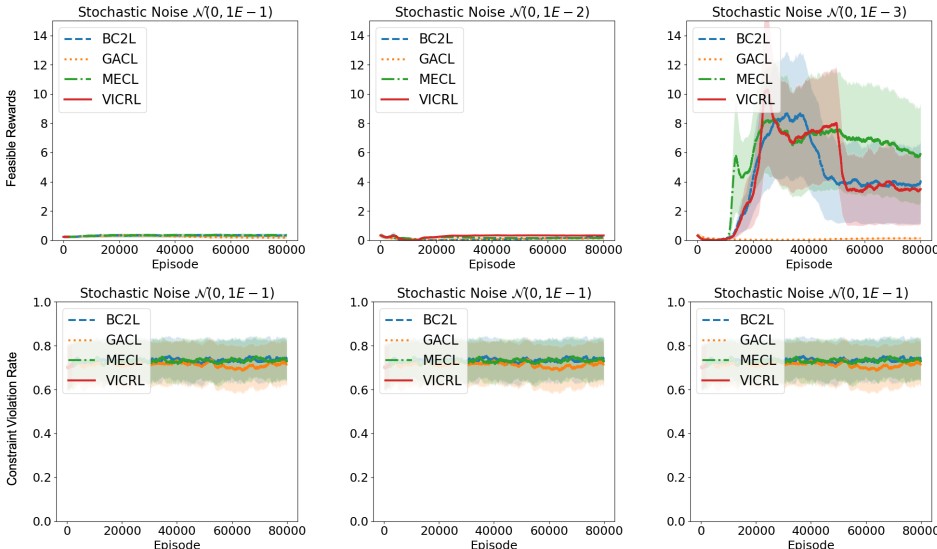

Figure D.7: From left to right, the transition function has the noises $\mathcal{N}(0, 0.001), \mathcal{N}(0, 0.01)$, and $\mathcal{N}(0, 0.1)$. The environment from top to bottom is Blocked Ant, Blocked Walker, Blocked Swimmer and Biased Pendulum. We use feasible rewards(top) and constraint violation rate(bottom) as the two metrics of the experiment.

