# OpenReview forum: "Benchmarking Constraint Inference in Inverse Reinforcement Learning"
_ICLR.cc/2023/Conference — ICLR 2023 poster_

### Official Review · Reviewer_4BCv · 2022-10-22

**Confidence:** 4
**Correctness:** 4
**Technical Novelty And Significance:** 2
**Empirical Novelty And Significance:** 2
**Recommendation:** 5

**Clarity, Quality, Novelty And Reproducibility:**

Most of my comments are above, but pointing out a few specific typos here to be fixed:

* Figure 3 caption defined both above and below figure?
* A few stray ICLR baselines instead of ICRL baselines :)
* Figure 7 both columns labeled “incomplete dataset” when one should be about added noise I think.


**Strength And Weaknesses:**

Strengths
* Clarity of presentation. With the exception of some minor typos here and there, some of which I point out below, the paper is well written and clear. The benchmark is well motivated from the perspective of a key missing piece for comparing different approaches to ICRL, and the method description was straightforward to follow.

* Potential for impact. The proposed benchmark seems like it would meaningfully fill a current gap to make reproducibility and comparisons across different methods for constraint inference easier. While I have some concerns that the benchmark as defined is too limited for meaningfully advancing the field as a whole, I believe in the general idea and think that if done well it could have a significant impact.

* Baseline comparisons. The baselines compared to in this paper are up-to-date methods that highlight different approaches to learning and inferring constraints from demonstrations. This aids in the interpretation of the results for the proposed VICRL method.

* Code looks reasonably straightforward to use with clear README which will be valuable for anyone interested in actually using the benchmark.

Weaknesses
* The VICRL method does not statistically outperform either of the presented state-of-the-art baselines across tasks (GACL or MECL). While it may perform comparably to prior techniques, there is no environment for which it statistically significantly outperforms baselines (based on Figures and Table B.2 of appendix). In order to then still show value of VICRL over the baselines, more information is needed as to where VICRL shines by having a posterior distribution over constraints. While what it learns appears to be interpretable based on the appendix, it is not obvious that the posterior distribution helps with this. So what exactly is the variational method providing in terms of interpretability, efficiency, or performance? If there is no clear answer to this, the paper might be better written as a benchmark only paper, with some extensions to the considered tasks.

* Proposed benchmark is lacking constraint and environment representations that the community already thinks are critical.
  * Grid-world representations not included, despite being mentioned by the authors in several places as a major focus of study for a subset of the RL community.
  * Constraints specified mostly as location constraints do not cover the vast varieties of constraint types that matter for robotics and RL applications
  * Many constraints that are interesting from a robotics standpoint include multiple variables, and sometimes multiple objects, and are often time-varying.
  * For example, consider those that describe the dynamics inside common simulators like PyBullet or MuJoCo, we have constraints like fixed (two bodies move together as one), revolute (fixed axis of rotation around which an object can move), or sliding. In safety applications, constraints are very likely to be time-varying, such as avoiding coming into contact with any humans.
  * None of these are represented by the current benchmark. This risks developing methods that are overly specific to static location-based constraints, which would not reflect what the community ultimately wants for downstream applications.

Not a weakness per se, but I didn’t understand how it could be possible that the constraint inference methods could out-perform PPO-Lag for the blocked ant environment. Could the authors elaborate on this?


**Summary Of The Paper:**

This paper makes two contributions. First, it proposes a new benchmark for Inverse Constrained Reinforcement Learning (ICRL) for continuous control tasks that provides environments along with location constraints in those environments that an agent has to satisfy, as well as expert demonstrations obtained through PPO-Lag for each of these. The benchmark includes continuous control tasks in MuJoCo and separately a continuous driving control task. Second, the paper proposes a new method, Variational Inverse Constrained Reinforcement Learning (VICRL) which uses variational inference to infer a full posterior over possible constraints. The paper compares VICRL to previously proposed constraint inference methods and finds that VICRL slightly outperforms the alternatives.

**Summary Of The Review:**

It is risky to write a paper that makes two contributions simultaneously. Doing so risks that neither component alone will be done sufficiently well to merit publication. Unfortunately I think that is the case with this paper. I personally think the introduction of the benchmark is likely to have a greater impact on the community than the VICRL technique, as the VICRL method does not statistically outperform either of the two state-of-the-art baselines for the presented tasks. However, as mentioned in the weaknesses section, the benchmark is not quite general enough to be used as a comprehensive assessment of agents’ abilities to learn constraints from demonstrations. I therefore do not think the paper can be accepted as is, and would encourage the authors to expand the benchmark part of their paper to consider a greater range in the types of constraints that are investigated.

---

> ### Author Response · Authors · 2022-11-13
> **Author response to the reviewer 4BCv - part 1**
>
> Dear Reviewer, thanks for your constructive comments. We have seriously considered your suggestions, and hopefully, the following response can address your concerns:
>
> - *"The VICRL method does not statistically outperform either of the presented state-of-the-art baselines across tasks (GACL or MECL). While it may perform comparably to prior techniques, there is no environment for which it statistically significantly outperforms baselines (based on Figures and Table B.2 of appendix). In order to then still show value of VICRL over the baselines, more information is needed as to where VICRL shines by having a posterior distribution over constraints. While what it learns appears to be interpretable based on the appendix, it is not obvious that the posterior distribution helps with this. So what exactly is the variational method providing in terms of interpretability, efficiency, or performance? If there is no clear answer to this, the paper might be better written as a benchmark only paper, with some extensions to the considered tasks."*
>
> **Our response:** Thanks for raising these concerns. In terms of positioning our paper, we agree the major contribution is about the benchmark, and this is explained by the paper title and the structure of the paper. Although we propose a VICRL extension, we treat it as one of the baselines implemented in our benchmark instead of a major methodological contribution. This is why we covered the details of VICRL with less than one page and used the rest of the paper for describing our benchmark. \textcolor{blue}{We will clarify this in the revised version.}
>
> When it comes to the contribution of VICRL, we argue VICRL is a strong baseline from the following perspectives:
>
> 1. VICRL is the first method that can estimate the posterior distributions of constraint functions. Based on these distributions, future works can explore the design of the probabilistic constraints or risk-aware constraint estimations (by computing Var or CVaR). Without VICRL, there is no baseline within the framework of Bayesian ICLR for continuous domains.
>
> 2. We argue the improvement of empirical performance is significant by considering a)  VCIRL outperforms all previous methods in 4 out of 7 tasks in the environments, although a more ideal case is to empirically outperform in all tasks.
> b) GACL or MECL can beat VICRL in one or two environments, but their advantage is inconsistent. For example, in the HighD environment, GACL can not learn any useful policy. For supporting our claims, we have added a significance test in Tables 2 and 3 on page 3.
>
> - *"Grid-world representations not included, despite being mentioned by the authors in several places as a major focus of study for a subset of the RL community."*
>
> **Our response:** Thanks for the suggestion.  In fact, grid-world environments are the ones we initially started with, but we decided to exclude them from our benchmark because of the following concerns:
>
> 1) **Practical reasons**. One of the main goals of proposing this benchmark is to fill the gap between the simulated environment with the real-world application (see our introduction). We agree Grid-world environments are the perfect test bed for studying some theoretical properties of proposed algorithms (it is easier to prove RL theorems under the tabular setting), but generalizing the performance of the discrete benchmark to real-world applications in continuous domain input is difficult.
>
> 2) **Theoretical reasons.** Under the discrete environments (e.g., Grid-worlds), Bellman Optimality can be reached by dynamic programming (an optimal policy ( $\pi^*=argmax_{a}Q^*(s,a)$ ) can be uniquely found), whereas, in this work, our baselines utilize function approximation for tackling continuous domains. These algorithms approximate the optimal policies by minimizing a sub-optimal upper bound (e.g., on the regret $\|\hat{Q}-Q^{*}\|_{\infty}$). There is no guarantee that our baseline can converge to the optimal policy satisfying the Bellman Optimality.  The gap between continuous space and tractable discrete space is significant, which induces a tradition to evaluate the tabular-based algorithms with Grid-worlds and the function approximation algorithms with continuous environments respectively.
>
> 3) **Novelty:** Please note that novelty is important in the creation of a benchmark, but there are already several Grid-Worlds benchmarks (see [4]).  We are surprised by the request to create more Grid-Worlds when Grid-Worlds are restricted to discrete states and are rarely realistic environments. Instead, the goal of a new benchmark is to pull the community towards new challenges such as continuous environments, and more realistic tasks.
>
> [4] Dexter R. R. Scobee and S. Shankar Sastry. Maximum likelihood constraint inference for inverse reinforcement learning. In 8th International Conference on Learning Representations, (ICLR).
> OpenReview.net, 2020.
>
> We have clarified these important concerns on Page 4.

---

> > ### Author Response · Authors · 2022-11-13
> > **Author response to the reviewer 4BCv - part 2**
> >
> > - *"Constraints specified mostly as location constraints do not cover the vast varieties of constraint types that matter for robotics and RL applications. "*
> >
> > **Our response:** Thanks for raising the concerns. Our benchmark contains two sets of environments. We agree the virtual environments focus on location constraints, however, our HighD (realistic) environments explored other types of constraints including vehicle distance and velocity.
> >
> > - *"Many constraints that are interesting from a robotics standpoint include multiple variables, and sometimes multiple objects, and are often time-varying. For example, consider those that describe the dynamics inside common simulators like PyBullet or MuJoCo, we have constraints like fixed (two bodies move together as one), revolute (fixed axis of rotation around which an object can move), or sliding."*
> >
> > **Our response:** Thanks for mentioning the constraints like fixed, revolute, and slides. When we look into their implementation in the simulators like PyBullet or MuJoCo, we realize these constraints are in fact the mechanical rules that the robot must follow. In other words, these constraints can not be violated in the simulator (or the simulator has not modeled the results or consequences when these rules are broken). We, therefore, argue these constraints (or rules) are not the appropriate targets of the prevailing online ICRL algorithms since we cannot generate nominal trajectories (i.e., trajectories with constraint-breaking events).
> >
> > - *" In safety applications, constraints are very likely to be time-varying, such as avoiding coming into contact with any humans.
> > None of these are represented by the current benchmark. This risks developing methods that are overly specific to static location-based constraints, which would not reflect what the community ultimately wants for downstream applications."*
> >
> > **Our response:** Thanks for suggesting the "avoiding human" constraints. In fact, one of the main constraints studied in our HighD environment is car distance, i.e., the vehicles must maintain a sufficient distance in order to prevent collision. This is similar to the "avoiding human" behavior in robots. Since cars drive very fast on German highways, preventing collisions is even more difficult.
> >
> > - *"Not a weakness per se, but I didn’t understand how it could be possible that the constraint inference methods could out-perform PPO-Lag for the blocked ant environment. Could the authors elaborate on this?"*
> >
> > **Our response:** We also want to clarify that although our benchmark uses PPO-Lag to generate expert demonstrations, PPO-Lag is **not** optimal in terms of maximizing the rewards or satisfying constraints. For the ICRL algorithms, since the agent can interact with the environment and receive feedback, its policy could outperform PPO-Lag in terms of collecting rewards. Most of the time, this improvement is accompanied by a higher constraint violation rate, but we do observe in some cases the agent can collect more rewards with a constraint violation rate that is similar to PPO-Lag's violation rate. When we observe the training process, we find ICRL agent conducts a larger scale of exploration at the beginning of training since the initialized constraint function is very soft (i.e., the distribution is flat with a large uncertainty), and thus it can not dominate the agent's preference. This exploration provides more knowledge to the agent compared to the PPO-Lag agent applying the hard constraint.

---

### Official Review · Reviewer_sTb2 · 2022-10-24

**Confidence:** 3
**Correctness:** 4
**Technical Novelty And Significance:** 2
**Empirical Novelty And Significance:** 3
**Recommendation:** 6

**Clarity, Quality, Novelty And Reproducibility:**

The paper is reasonable well written and collected together several existing domains to construct the benchmark. The algorithm appears to be a straightforward adaptation of ideas from Scobee et al (as cited) to include a distribution on the constraints.

**Strength And Weaknesses:**

# Strength

1. I agree with the paper that this (important) emerging sub-area is in need of common benchmarks.

2. The domains and proposed constraints/demonstrations are certainly interest and cover both synthetic and "real world" scenarios.

3. The proposed algorithm makes sense and builds on prior work in a pretty reasonable way.

# Weakness

As I tried to highlight in the summary, this benchmark focuses on domains that are entirely deterministic (the synthetic ones) or approximately deterministic (the ones generated by human performance noise). Like with the development of maximum entropy inverse RL, it is worth distinguishing this from fundamentally stochastic domains, i.e., when causal entropy is not well approximated by "non-causal" entropy. Here I would argue that the gridworld domains that were directly disregarded offer a non-trivial test bed, e.g., for high-level planning in the presence of environment and multi-agent noise.

In particular, the proposed algorithm is really only suitable for (approximately) deterministic domains as is pointed out in Brian Ziebart's 2010 thesis.

[Nitpick] Finally, I would argue that the proposed algorithm addresses a slightly different problem since it is privy to an implicit prior distribution over constraints?



**Summary Of The Paper:**

This paper can be thought of as two (related) papers:

1. A proposed benchmark for learning constraints from demonstrations in continuous (approximately deterministic) domains. Here constraint is taken to be an assertion that sum of unknown random variables (either in expectation or per episode) is less than a particular value. The goal of this family of problems is to identify the unknown constraints. This includes a series of domains with corresponding demonstrations, objectives, and constraints.

2. An adaptation of maximum entropy based constraint inference that explicitly models distributions over the constraints. The adaptation is shown to perform well on the proposed benchmark making it a reasonable baseline for future comparisons.



**Summary Of The Review:**

I think the area of constraint learning is indeed in need of a set of benchmarks. I think this paper is a good step in that direction, although I would argue has a fairly biased focus on continuous deterministic domains that is not emphasized enough as a limitation / bias.

---

> ### Author Response · Authors · 2022-11-13
> **Author response to the reviewer sTb2 - Part 1**
>
> Dear Reviewer, thanks for your constructive comments. We have seriously considered your suggestions, and hopefully, the following response can address your concerns:
>
> - *"As I tried to highlight in the summary, this benchmark focuses on domains that are entirely deterministic (the synthetic ones) or approximately deterministic (the ones generated by human performance noise). "*
>
> **Our response:** Thanks for raising the concerns. Our benchmark mainly contains two sets of environments. We confirm that the initial versions of MuJoCo environments are deterministic (but we have extended them to the stochastic environment in the updated version, see Figure 6 on Page 9.), but the HighD environments are stochastic. The stochastic game dynamics are due to the aleatoric uncertainty induced by stochastic human driving.
>
> 1) The stochastic human-driving trajectories. These trajectories reflect human preference under different road conditions.  Different drivers in the exact same situation will drive differently, yielding different trajectories. Hence, the population of drivers induces underlying transition dynamics that are stochastic.  The trajectories in the HighD dataset are essentially samples from these stochastic transition dynamics, based on which we build the HighD environments. Each time an environment is reset (either the game ends or the step limit is reached), it randomly picks a scenario with a set of driving trajectories. This is equivalent to sampling from the aforementioned transition dynamics. Since the game states $s$ are constructed based on these trajectories (see Figure 2), the agents will observe different $s^{\prime}$ after performing an action $a$ under a state $s$ across different episodes. The distribution of $s^{\prime}$ essentially reflects the stochasticity in the underlying dynamics.
>
> 2) The amount of stochasticity induced by human drivers is unknown since the underlying distribution is unknown.  Whether this environment is nearly deterministic or not is beside the point since the goal is not to design environments with as much noise as possible, but to design realistic environments with realistic noise.
>
> We have expanded the descriptions in our paper to prevent further misunderstandings (see Page 4).
>
> - *" Like with the development of maximum entropy inverse RL, it is worth distinguishing this from fundamentally stochastic domains, i.e., when causal entropy is not well approximated by "non-causal" entropy. In particular, the proposed algorithm is really only suitable for (approximately) deterministic domains as is pointed out in Brian Ziebart's 2010 thesis."*
>
> **Our response:** Thanks for raising the concerns regarding the deterministic domains. To answer this question, we extend the MuJoCo environment by including Gaussian noise in the transition dynamics and show the results in Figure 6. we find ICRL models are generally robust to additive Gaussian noises in environment dynamics until they reach a limit (e.g., N (0, 0.1)). In fact,  the deterministic constraint inference methods (MECL and B2CL) can benefit from a proper scale of random noise since these noisy signals facilitate a more restricted constraint function and thus a lower constraint violation rate.
>
> We also appreciate the reviewer for mentioning **causal entropy**. We assume the causal entropy refers to $\mathcal{H}(A_{0:t}\|S_{0:t})$ [1]. We confirm that our current formulation of ICRL is trajectory-oriented (instead of time-oriented), which naturally models non-causal entropy $\mathcal{H}({\pi})$ without depending on transition dynamics, and unfortunately, we are unaware of any causal-entropy-based constraint inference algorithms, especially for the ones based on the neural function approximation. How to extend the causal entropy methods (i.e., inverse soft-q learning) for constraint inference will be a promising future direction.
>
> [1] Michael Bloem and Nicholas Bambos. Infinite Time Horizon Maximum Causal Entropy Inverse Reinforcement Learning. IEEE Conference on Decision and Control, (CDC), 2014.

---

> > ### Author Response · Authors · 2022-11-13
> > **Author response to the reviewer sTb2 - Part 2**
> >
> > - *"Here I would argue that the grid world domains that were directly disregarded offer a non-trivial test bed, e.g., for high-level planning in the presence of environment and multi-agent noise."*
> >
> > **Our response:** Thanks for suggesting the grid-world domains. In fact, grid-world environments are the ones we initially started with, but we decided to exclude them from our benchmark because of the following concerns.
> >
> > 1) **Practical reasons**. One of the main goals of proposing this benchmark is to fill the gap between the simulated environment with the real-world application (see our introduction). We agree Grid-world environments are the perfect test bed for studying some theoretical properties of proposed algorithms (it is easier to prove RL theorems under the tabular setting), but generalizing the performance of the discrete benchmark to real-world applications in continuous domain input is difficult.
> >
> > 2) **Theoretical reasons.** Under the discrete environments (e.g., Grid-worlds), Bellman Optimality can be reached by dynamic programming (an optimal policy ( $\pi^*=argmax_{a}Q^*(s,a)$ ) can be uniquely found), whereas, in this work, our baselines utilize function approximation for tackling continuous domains. These algorithms approximate the optimal policies by minimizing a sub-optimal upper bound (e.g., on the regret $\|\hat{Q}-Q^{*}\|_{\infty}$). There is no guarantee that our baseline can converge to the optimal policy satisfying the Bellman Optimality.  The gap between continuous space and tractable discrete space is significant, which induces a tradition to evaluate the tabular-based algorithms with Grid-worlds[2]] and the function approximation algorithms with continuous environments respectively [3].
> >
> > [2] Dexter R. R. Scobee and S. Shankar Sastry. Maximum Likelihood Constraint Inference for Inverse Reinforcement Learning. International Conference on Learning Representations, (ICLR), 2020
> >
> > [3] Shehryar Malik, Usman Anwar, Alireza Aghasi, and Ali Ahmed. Inverse constrained reinforcement
> > learning. In Proceedings of the 38th International Conference on Machine Learning, ICML 2021.
> >
> > - *" [Nitpick] Finally, I would argue that the proposed algorithm addresses a slightly different problem since it is privy to an implicit prior distribution over constraints?"*
> >
> > **Our response:** Thanks for raising this concern. We agree that our VICRL algorithm enables assigning different priors over the constraint while previous algorithms do not. To simplify the problem, we treat the Beta prior as hyper-parameters and determine their values experimentally, but we can also use other priors (e.g., the informative prior on human preference as mentioned above) which might further improve the model performance. We leave this to future work.

---

### Official Review · Reviewer_cYaH · 2022-10-25

**Confidence:** 4
**Correctness:** 3
**Technical Novelty And Significance:** 3
**Empirical Novelty And Significance:** 2
**Recommendation:** 6

**Clarity, Quality, Novelty And Reproducibility:**

Clarity & Quality: The manuscript is quite clear and high quality.

Novelty:  The bayesian approach by the authors is fairly novel, and the subfield targeted by this manuscript is clearly in need of some kind of rallying "benchmark".

Reproducibility: Code has not been shared, but it appears that it will be once deanonymization occurs.

**Strength And Weaknesses:**

Strengths: The benchmark seems well motivated, and the experiments provided by the authors seem sufficient to establish the shape of the problem, but also to gesture at headroom for improvement.

Weaknesses: It's unclear to me how long this will remain a relevant benchmark.  The MuJoCo suite is a bit dated now, and the community has tended to overfit to it as a reference simulator.  A benchmark like the one proposed by the authors, but with significantly more variability / range of difficulty seems like it would be strictly more useful, and possibly last longer as a concrete target for the community.  As it stands, I worry that this benchmark will be "solved" within approximately 6 months.

**Summary Of The Paper:**

The authors provide a new benchmark for constrained inverse reinforcement learning, including modified mujoco environments, as well as a self-driving-car inspired environment.  The authors additionally propose a Bayesian algorithm for solving problems in this class they call Variational Inverse Constrained Reinforcement Learning (VICRL).

**Summary Of The Review:**

A solid submission, including a new algorithmic technique and a new benchmark.  I have some reservations about lasting impact, but these are not show-stopping reservations.

---

> ### Author Response · Authors · 2022-11-13
> **Author response to the reviewer cYaH**
>
> Dear Reviewer, thanks for your constructive comments. We have seriously considered your suggestions, and hopefully, the following response can address your concerns:
>
> *"Weaknesses: It's unclear to me how long this will remain a relevant benchmark. The MuJoCo suite is a bit dated now, and the community has tended to overfit to it as a reference simulator. A benchmark like the one proposed by the authors, but with significantly more variability / range of difficulty seems like it would be strictly more useful, and possibly last longer as a concrete target for the community. As it stands, I worry that this benchmark will be "solved" within approximately 6 months."*
>
> **Our response:** Thanks for raising the concerns. We agree that a strong benchmark should be challenging and remain effective for a reasonable amount of time, which we clarify from the following perspectives.  Note also that the age of an engine such as MuJoCo should not be a reason in itself to determine whether environments based on MuJoCo are worthy or not, but whether those environments are challenging.  Note also that we are not proposing to use the popular control tasks based on MuJoCo as they are.
>
> 1) MuJoCo was originally proposed for evaluating control algorithms, but in this work, we extend the MuJoCo simulator to test Inverse Constrained Reinforcement Learning (ICRL) algorithms, which is an emerging research topic without any common benchmark. ICRL is more difficult than RL since we must learn both an imitation policy and the corresponding constraint functions from the demonstration dataset and the environment. We also find there is a limited number of works that solve ICRL under continuous domains.
>
> 2) Our benchmark includes a HighD environment, which is a non-episodic and stochastic environment.  We realize some algorithms (e.g., GACL) that perform well in the deterministic MuJoCo environment can not learn any satisfying policy or constraint in HighD. The gap between the performance of popular baselines and the corresponding upper/lower bounds (for rewards and violation rate) is big. The space for improvement is significant.
>
> 3) In order to increase variability or difficulty, we included tasks in the benchmarks  for a) inferring two different constraints (velocity and car distance) from the HighD environment (see Multi-Constraints Environment on Page 9) b) Inferring constraints from sub-optimal demonstration dataset (See "Noisy Demonstration" on Page 9) and c) Inferring constraints from incomplete demonstration dataset (See "incomplete Demonstration" on Page 9).
>
>
> 4) In addition, our ICRL benchmark enables incorporating any external
> constraints into different environments. Our code provides an **interface that allows editing ground-truth constraints in the environment**. After the edit, the user needs to generate a demonstration dataset by training an expert agent (see Section 4.3). During this process, the most important step is demonstrating that the added constraint is effective (See Section 4.4), and this is how we empirically selected the constraints in the paper. We encourage users to experiment with other constraints by using our interface. In the future, more combinations can be added and there is plenty of space to increase both difficulty and variability.
>
> We summarize the major challenges in our benchmark (see Page 10). Appendix C further elaborates on the limitation, challenges, and open questions in our benchmark.

---

### Author Response · Authors · 2022-11-13
**A Summary of Updates**

Dear Reviewers, Area Chairs, and Program Chairs,

We are greatly thankful for the insightful comments and suggestions, which are very helpful for us to further improve this work. We have added clarifications, explanations, and additional experiment results to our paper (marked by the **blue** color). To clarify our modifications and prevent misunderstanding, we summarize our major updates in the following:

1. **Extending MuJoCo to Stochastic Domain**. We have improved the virtual environments by implementing a stochastic transition function in the MuJoCo environments. Since MuJoCo is an opened-source game engine, it enables modifying the transition function by adding Gaussian noise to the generated game states and utilizing these states to simulate the next scenario for controlling. We have explored the options of adding different scales of noise to the environment and testing the corresponding performance of each ICRL algorithm. The results show that ICLR baselines are generally robust to random noise. See Figure 6 and Page 9 in the modified version.

2. **Clarifying the Stochacity in the HighD environment.**
The HighD environment is constructed based on the samples from stochastic dynamics in the human driving environment.  We have clarified how the dynamics in HighD can reflect the stochastic human preference and why HighD should be viewed as a stochastic environment (See Page 5).

3. **A Summary of Majors Challenges**. As it is requested by the reviewer cYaH, we have summarized the major challenges in our benchmark (see Page 10 and Appendix C) from the perspectives of constraint satisfaction, noisy demonstration data, and High-dimensional Features. We believe these challenges are not trivial and will inspire the following works.

4. **Significance Test**. In order to examine whether the results from our VICRL models are significantly different from other baselines (for addressing the concerns from the reviewer 4BCv), we conduct a Wilcoxon signed-rank test and report the results (including the averaged p values) in Table 2 and Table 3.

5. **Discussion about the Gird World Environment**. We have clarified the major reasons why our benchmark focuses on the continuous domain and why the Gird-World environment is not included from practical and theoretical perspectives (See Page 4).

Apart from the academic contributions in the paper, our benchmark has **some realistic values**. The development of this benchmark is based on the demand of our industrial partners, who requested developing techniques to infer constraints on speed and car distance.  Our industrial partners plan to incorporate such constraints in their RL development stack (for supporting autonomous driving agents), so they want to know how the ICRL algorithms perform in some challenging domains that can more or less resemble realistic applications. The popular benchmarks (e.g., Grid-World and MuJoCo) are not satisfying, and this is why we extend them by **adding new constraints and complex environments** (e.g., highway driving tasks). **Besides, our benchmark provides an interface for including other constraints of interest in the environment**. We believe our benchmark could be a stepping stone in facilitating the development of more mature ICRL algorithms.

---

> ### Author Response · Authors · 2022-11-26
> **Any further questions or comments?**
>
> Dear Reviewers,
>
> We want to thank you for your encouraging comments. We have addressed your questions in the response and incorporated them in the revised manuscript.
>
> Please let us know if you have any more questions, and we would be happy to address them within our allowed period.
>
> Warmest regards,
> Authors of Paper 2680.

---

### Decision · Program_Chairs · 2023-01-20

**Decision:**

Accept: poster

**Justification For Why Not Higher Score:**

Concerns regarding the longevity of the benchmark.

**Justification For Why Not Lower Score:**

Solid contribution and strong author rebuttal.

**Metareview: Summary, Strengths And Weaknesses:**

I thank the authors for their submission and active engagement during the discussion period. This is a borderline paper. On the positive side, reviewers found the work to be novel [cYaH], the paper clear [cYaH, 4BCv], the problem important [sTb2, 4BCv], the domains used for evaluation interesting [sTb2], the proposed approach reasonable [sTb2] and the comparison to baselines reasonable [4BCv]. On the negative side, reviewers had concerns around deterministic nature of the environments [sTb2], underwhelming improvements over baselines [4BCv], and concerns regarding the longevity of the benchmark [cYaH]. However, I believe the authors have addressed these concerns (particularly regarding determinism of the simulator) well in their rebuttal. Therefore, I recommend acceptance but highly encourage the authors to take the reviewer feedback into account for the camera ready version of the paper.

**Note From Pc:**

if the above contains the word "oral" or "spotlight" please see: "oral" presentation means -> notable-top-5% and "spotlight" means -> notable-top-25%. As stated in our emails, we are disassociating presentation type from AC recommendations